# CMQUANT: A QUANTIZATION-AWARE PARAMETER-EFFICIENT FINE-TUNING FRAMEWORK FOR 4-BIT CONSISTENCY MODELS

## ABSTRACT

Consistency Models (CMs), built on diffusion models, use model state trajectory fitting to reduce iterations required for sample generation. However, they still maintain high per-iteration computational costs and large model parameter sizes, which hinder deployment on resource-constrained devices. Quantization, an effective model compression technique with notable success in Large Language Models, remains largely unexplored for CMs. We observe that the unique characteristics of CMs pose significant obstacles to effective quantization. First, the trajectory fitting errors inherent to CMs accumulate across iterations and are further amplified when quantization errors are involved. Second, CMs training often relies on Low-Rank Adaptation (LoRA), which injects low-rank matrices into specific layers without altering pretrained weights. However, quantization errors not only disrupt this initialization consistency, but also hinder training optimization and impair convergence. In this paper, we propose CMQuant, a novel quantization-aware parameter-efficient fine-tuning framework tailored for CMs. CMQuant introduces three innovations: (1) Trajectory Distillation with Phased Targets (TDPT), which assigns distinct optimization objectives to different stages of trajectory, enables accurate starting points for each stage and thereby minimizes accumulated quantization errors in iterations. (2) Hessian-Guided SVD-Initialized LoRA (HGS-LoRA), which leverages hessian-guided matrix decomposition to initialize LoRA, directing weight updates along quantization-friendly paths and thereby reducing quantization errors. (3) Quantization-Aware Rank Adaptation (QRA), which assigns ranks adaptively based on the degree of variation in activations and weights across different CMs layers. This minimizes the impact of quantization without increasing the total number of LoRA parameters. By integrating quantization into the CMs training, CMQuant achieves the first 4-bit quantization of CMs for both weights and activations. Experiments show that CMQuant outperforms SOTA at least FID↓/PickScore↑/IC↑ of 4.74/11.61/3.01 on FLUX. Furthermore, it improves throughput by 1.71×/3.43× on SDXL/FLUX, with only 27%/25% memory footprint.

## 1 INTRODUCTION

Consistency models (CMs) Luo et al. (2023a); Wang et al. (2025); Zheng et al. (2024a); Wang et al. (2024); Ren et al. (2024), built on diffusion models Sohl-Dickstein et al. (2015), deliver significant inference speedups by reducing the number of iterations needed to generate samples. This efficiency arises from their core mechanism: training enforces a self-consistency property where all points along the same probability flow Ordinary Differential Equation (PF-ODE) Song et al. (2023) trajectory map to an identical solution point. However, CMs do not reduce either the model parameter sizes or the computational cost per iteration. For instance, CMs built on the FLUX Labs (2024) require approximately 23GB of memory under FP16 precision and 14.23 TFLOPS of computation per iteration. Such high resource demands severely hinder deployment on resource-constrained devices.

Quantization He et al. (2023); Chen et al. (2024); Ke et al. (2025) has emerged as a promising technique for reducing computational costs and parameter sizes by converting model weights and activations into lower-bit fixed-point representations. While it has achieved notable success in

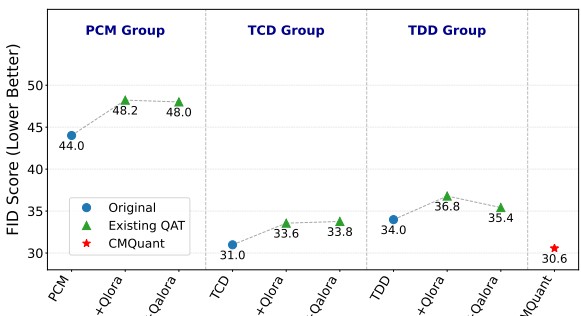

Figure 1: In terms of FID (lower is better), existing QAT methods are detrimental to the performance of CMs.

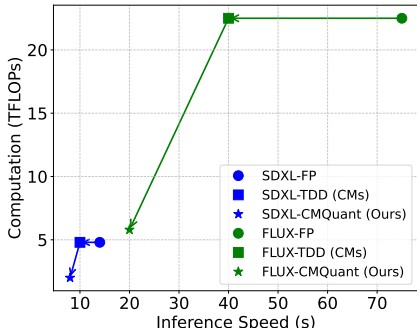

Figure 2: Computation costs and inference speed of different methods for SDXL and FLUX.

Large Language Models Hu et al. (2025); Xu et al. (2025b), its application to CMs remains largely underexplored. We observe that directly applying quantization to CMs faces three key challenges: (1) To improve the accuracy of PF-ODE trajectory fitting, most existing CMs decompose the trajectory into multiple sub-trajectories and then fit each one sub-trajectory independently Wang et al. (2024; 2025). However, since the output of the consistency model's sub-trajectory is an approximation of the full diffusion process, the deviations (i.e., approximation error) propagate and compound across sub-trajectories, leading to an increasingly larger gap from the the original PF-ODE trajectory. Worse still, the problem becomes more severe when quantization is involved: quantization error itself degrades the fitting precision of each sub-trajectory, and when this error combines with the pre-existing compounded deviations, it creates a synergistic negative effect that further widens the gap from the original PF-ODE trajectory. (2) The training of CMs often relies on Low-Rank Adaptation (LoRA) Hu et al. (2022), whose initialization involves injecting low-rank matrices into specific layers without modifying the original pretrained weights. However, quantization disrupts this equivalent consistency in the initialization process and interferes with optimization during training, thus impairing the training convergence of CMs. (3) The activation distributions in different layers exhibit pronounced variations throughout the iteration process. A similar phenomenon is observed with the weight distributions across these layers. Unfortunately, existing LoRA approaches fail to account for these inherent layerwise differences in CMs, and assign same rank to all layers. What is more concerning, the problem becomes more prominent when quantization is incorporated: quantization errors across different layers exhibit significant discrepancies, which further exacerbates the differences between layers and thus undermines the reasonableness of treating all layers equally.

To address these challenges, we propose CMQuant, a novel Quantization-Aware Parameter-Efficient Fine-Tuning (PEFT) Xu et al. (2023) framework tailored for CMs. CMQuant comprises three core components: (1) Trajectory Distillation with Adaptive Targets (TDAT): Unlike previous methods that adopt the same optimization objective for all sub-trajectories, our approach assigns distinct targets to different sub-trajectories. This strategy, which fully accounts for the iterative inference characteristics of CMs, helps achieve accurate starting points for each early sub-trajectory, thereby mitigating the accumulation of errors across iterations; (2) Hessian-Guided SVD-Initialized LoRA (HGS-LoRA): Using SVD decomposition guided by the Hessian matrix of activations, we freeze the main components within weights that impact quantization, while using the remaining components as the initialization weights for LoRA. This suppresses the generation of large quantization errors and constrains model updates to quantization-friendly directions, which further alleviates the impact of quantization on CM training; (3) Quantization-Aware Rank Adaptation (QRA): Adaptive assignment of LoRA ranks according to activation variations across iterations and weight change rates across training steps, QRA minimizes the impact of quantization without increasing the total number of LoRA parameters. With these three modules mitigating quantization errors, CMQuant subtly integrates quantization into the CM training process. CMQuant achieves the first 4-bit quantization of CMs for both weights and activations without compromising the performance of CMs.

Experiments demonstrate that CMQuant achieves SOTA performance on CMs (see Fig. 1 for detailed comparisons). Specifically, for the FLUX model, CMQuant outperforms existing approaches at least 4.74/11.61/3.01 on FID↓/PickScore↑/IC↑. Additionally, our method achieves 1.71×/3.43× higher throughput and 27%/25% memory savings compared to the baseline. Our main contributions are summarized as follows:

- We identify that the direct application of Quantization to CMs fails primarily due to two key limitations: accumulation of quantization errors across CMs inference iterations, and incompatibility between existing LoRA and quantization in CMs.

- We propose CMQuant, a novel Quantization-Aware PEFT framework. Composed of three core components (TDPT, HGS-LoRA, and QRA), this framework elegantly and effectively integrates QAT into the training of CMs.

- The performance of CMQuant is verified on popular generative models with 1.1B, 3.5B, and 16.7B parameters. It achieves considerable inference efficiency for CMs while preserving their generation quality and semantic matching capabilities.

## 2 RELATED WORK

**Diffusion Models.** Diffusion models (Yang et al., 2023) have attracted extensive attention for their potent image generation ability. Recent studies have shifted from convolution-based UNet architectures (Ronneberger et al., 2015; Ho et al., 2020) to transformer-based designs (Peebles & Xie, 2023; Bao et al., 2023) and scaled up model sizes (Esser et al.). During inference, diffusion models generate high-quality samples by iteratively denoising random noise into data points through a multi-step process. While effective, this iterative mechanism incurs substantial computational costs, severely restricting their use in real-time applications.

**Consistency Models.** Consistency models demonstrate promise in accelerating the sampling process by learning a direct mapping from noise to data based on the trajectory of the Probability Flow Ordinary Differential Equation (ODE). This approach reduces the number of iterations while maintaining image quality. LCMs (Luo et al., 2023a) and LCM-LoRA (Luo et al., 2023b) combine consistency distillation with latent diffusion models Rombach et al. (2022), achieving remarkable performance in accelerating text-to-image synthesis. Moreover, PCM Wang et al. (2024) simplify the learning process by decomposing the ODE trajectory into multiple sub-trajectories. Recent works such as Target-Driven Distillation Wang et al. (2025) and Trajectory Consistency Distillation Zheng et al. (2024a) further optimize this process by introducing multiple target time points, which enables anytime-to-anytime jumps during inference and enhances image quality. However, excessive target time points inevitably introduce an accumulation of errors in steps, which degrades the final output.

**Quantization-Aware Training with Lora.** Building on the foundation of LoRA Hu et al. (2022), QLoRA Dettmers et al. (2023) was the first to introduce a memory-efficient fine-tuning method. Its core design involves quantizing pre-trained models to low-bit representations and fine-tuning a high-precision LoRA component. This approach allows for effective fine-tuning of large language models even with limited memory resources. Taking this as a starting point, LoFTQ (Li et al., 2023b) and LQ-LoRA (Guo et al., 2023) initialize with the quantization loss from the original weights. QALoRA (He et al., 2023) perform groupwise fine-tuning to learn an additional high-precision group-wise bias for the quantized model. However, merging the low-bit pretrained model with its high-precision LoRA component still yields high-precision weights in the end, which fails to boost inference speed. Furthermore, most existing methods focus solely on weight quantization while neglecting activation quantization, leaving the model with substantial computational overhead.

## 3 PRELIMINARIES

**Consistency Models.** For a given data set $\{\mathbf{x}_t | t \in [0, T]\}$, the stochastic trajectory is defined as: $d\mathbf{x}_t = f(\mathbf{x}_t, t)dt + g(t)d\mathbf{w}_t$. With $\mathbf{w}_t$ represents standard Brownian motion, $f(\mathbf{x}_t, t)$ for the drift coefficient, and $g(t)$ the diffusion coefficient for stochastic variations. When $g(t)$ is set to zero, the diffusion process can be described by an Ordinary Differential Equation (ODE). This ODE, which has the same marginal distribution as the diffusion process, is called a Probabilistic Flow-Ordinary Differential Equation (PF-ODE). Instead of estimating the scores of marginal distributions, CMs directly predicts the solution points of the PF-ODE trajectories by enforcing the self-consistency property. Specifically, when $\mathbf{f_\theta}(\cdot, \epsilon) = \mathbf{x}_\epsilon$ ($\mathbf{x}_\epsilon$ denotes the target final output), the consistency model $\mathbf{f_\theta}(\cdot, t)$ learns to satisfy the condition $\mathbf{f_\theta}(\mathbf{x}_t, t) = \mathbf{x}_\epsilon$ by enforcing:

$$\mathbf{f_\theta}(\mathbf{x}_t, t) = \mathbf{f_\theta}(\mathbf{x}_{t'}, t') \quad \forall t, t' \in [\epsilon, T] \tag{1}$$

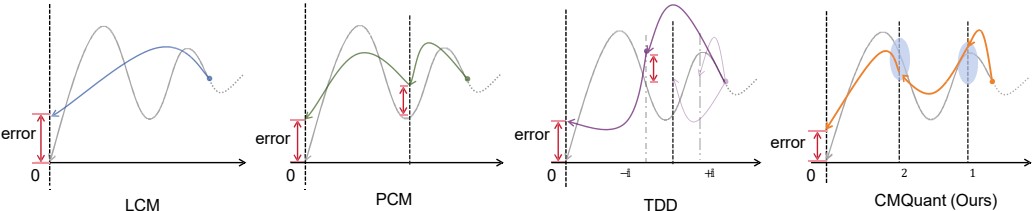

Figure 3: The gray line represents the original PF-ODE, while the colored lines represent the new ones fitted by different methods. For CMQuant, correcting the starting points of intermediate iterations mitigates error accumulation, resulting in a smaller final mean squared error.

**Quantization-Aware Training with LoRA.** Quantization involves mapping floating-point numbers to discrete intervals using integer values. The quantization process is defined as follows:

$$\mathcal{Q}(\boldsymbol{W}) = \text{clamp}\left(\left\lfloor \frac{\boldsymbol{W}}{s} \right\rceil, q_{\min}, q_{\max}\right) \tag{2}$$

Here, $\boldsymbol{W}$ denotes the weight matrix, $s$ represents the quantization step size, and $q_{\min}, q_{\max}$ specifies quantization bounds. In contrast to Post-Training Quantization (PTQ) Hu et al. (2025); Xu et al. (2025a;b) methods that directly quantize models using Eq. 2, Quantization-Aware Training (QAT) methods adjust weights through training to minimize the final quantization loss.

Low-rank adaptation decouples linear weights into the multiplication approximation of low-rank matrices. Udpate of weights $\Delta W$ is transferred to $\Delta A \Delta B$, where $W + \Delta W \approx W + \Delta A \Delta B$, and original parameter $W$ is frozen. The frozen weights are further quantized to reduce memory consumption during training, at the cost of introducing quantization error. QLoRA Dettmers et al. (2023) is a typical example of this approach. Its application to linear layers is expressed as follows:

$$\text{Linear}(X) = \mathcal{Q}(W)\, X + AB\, X \tag{3}$$

As indicated by the formula, although the original weights are quantized, the floating-point $A$ and $B$ still yield high-precision weights in the end. Moreover, the input $X$ is not quantized, resulting in no reduction in computational costs.

## 4 METHODOLOGY

### 4.1 TRAJECTORY DISTILLATION WITH PHASED TARGETS

Consistency models (CMs) enforce self-consistency to fit a new PF-ODE, which enable fast inference by reducing the number of iterations required for sample generation. However, the trajectory of this new PF-ODE inevitably shows deviations that undermine the performance of CMs, as illustrated in Fig. 3. Recent works such as PCM Wang et al. (2024) and TDD Wang et al. (2025) aim to mitigate the final deviations by splitting the PF-ODE trajectory into multiple sub-trajectories and fitting each individually. But these approaches introduce an additional issue: deviations in early sub-trajectories alter the input distribution for subsequent ones, causing these deviations to compound across the sub-trajectories and leading to an increasingly larger gap from the the original PF-ODE trajectory. Worse still, the problem becomes more severe when quantization is involved: quantization error itself degrades the fitting precision of each sub-trajectory, and when this error combines with the pre-existing compounded deviations, it creates a synergistic negative effect that further widens the gap from the original PF-ODE trajectory.

To alleviate this issue, we propose Trajectory Distillation with Phased Targets (TDPT). Like previous methods, TDPT splits the original PF-ODE into multiple sub-trajectories. The key difference is that we assign different fitting targets for each sub-trajectories. For sub-trajectories corresponding to the first few iterations of CMs, as shown in Eq. 4, we use the output of the untrained quantized model as the target. For others, we retain the output of the original floating-point model as the target. The variable definitions in the following formula are consistent with those in Eq. 1, and $\mathbf{f}_{\mathcal{Q}(\boldsymbol{\theta})}$ denotes untrained quantized CMs.

$$\mathbf{f}_{\boldsymbol{\theta}}(\mathbf{x}_t, t) = \begin{cases} \mathbf{f}_{\boldsymbol{\theta}}(\mathbf{x}'_t, t') & \forall t, t' \in [\epsilon, T'] \\ \mathbf{f}_{\mathcal{Q}(\boldsymbol{\theta})}(\mathbf{x}'_t, t') & \forall t, t' \in [T', T] \end{cases} \tag{4}$$

Quantization errors from untrained quantized CMs in a single iteration do not significantly degrade the performance of CMs. The primary factor affecting CMs' performance is the propagation and amplification of such errors across iterations, as detailed in Appendix A.4. The output of CMs at time $t$ serves as input at time $t - 1$. Inaccuracies in the input at time $t - 1$, which interact with quantization errors, introduce further inaccuracies into the output at $t - 1$. Thus, the accumulation and exacerbation of errors ultimately originate from inaccuracies in the output at time $t$. To mitigate this issue, it is crucial to ensure the accuracy of the output at time $t$. Specifically, during the PF-ODE trajectory fitting process, we must identify accurate starting points $\mathbf{f}_{\mathcal{Q}(\boldsymbol{\theta})}(\cdot, t)$. This insight leads us to use the output of untrained quantized CMs as the fitting target for earlier sub-trajectories. This strategy avoids forcing quantized CMs to fit potentially unattainable floating-point model outputs; instead, it directs them to fit outputs from untrained quantized CMs that lie close to these floating-point outputs. This ensures each sub-trajectory in this phase has an accurate starting point, thereby aligning inference with training. In other words, this design enhances the model's robustness against inaccuracies in starting points. Furthermore, since the PF-ODE trajectory of quantized CMs is not the one we truly aim to fit, we do not use the quantized output as the target for all sub-trajectories. Therefore, for the final sub-trajectory, we use the output of original floating-point model as the target. As detailed in the Appendix A.8, this design fully accounts for the iterative inference characteristics of CMs, mitigates the accumulation of errors across iterations.

### 4.2 HESSIAN-GUIDED SVD-INITIALIZED LOW-RANK ADAPTATION

The training of CMs often relies on Low-Rank Adaptation (LoRA) Hu et al. (2022), whose initialization involves injecting low-rank matrices into specific layers without modifying the original pretrained weights. However, quantization disrupts this equivalent consistency in the initialization process and interferes with optimization during training, thus impairing the training convergence of CMs.

To address these limitations, we propose Hessian-Guided SVD-Initialized Low-Rank Adaptation (HGS-LoRA). Using SVD guided by the Hessian matrix of activations, we freeze the main components within weights that impact quantization, while using the remaining components as the initialization weights for LoRA. This strategy suppresses the gen-

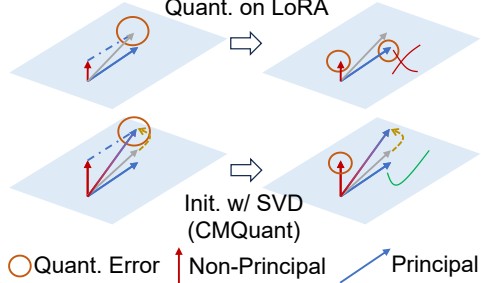

Figure 4: HGS-LoRA prevents the principal components from quantization error (4-bits) and continue to train on the non-principal components at high precision (16-bits).

eration of large quantization errors and constrains model updates to quantization-friendly directions, which further alleviates the impact of quantization on CM training.

We verify that quantization error is jointly determined by the Hessian matrix of activation statistics and the weights, as derived in Appendix A.2. The Hessian matrix measures the second-order sensitivity of the quantization loss function to changes in activations. By multiplying it with the weights, we reweight the latter to capture how each component within the weights actually contributes to quantization errors. Furthermore, we perform SVD on these reweighted weights to derive their principal components (major contributors to quantization error) and non-principal components (minor contributors). That is to say, errors in principal components would significantly amplify the final quantization error, whereas those in non-principal components have relatively minor effects. Therefore, we retain the principal components as untrainable weights to prevent changes that could cause large quantization errors, as shown in Eq. 5. The non-principal components are decomposed into LoRA adapter matrices to facilitate training of CMs while maintaining stability during initialization. Here, $E$ takes the $r$ largest singular values in $W$, and $H$ represents the Hessian matrix of activations.

$$W^+ = UEV H^{-\frac{1}{2}}, \quad AB = W - W^+, \quad \text{where } UEV = \text{SVD}\left(WH^{\frac{1}{2}}\right). \quad (5)$$

### 4.3 QUANTIZATION-AWARE RANK ADAPTATION

The activation change rates over iterations vary considerably across different layers of CMs, while weight distributions and training convergence rates also differ significantly. Unfortunately, existing

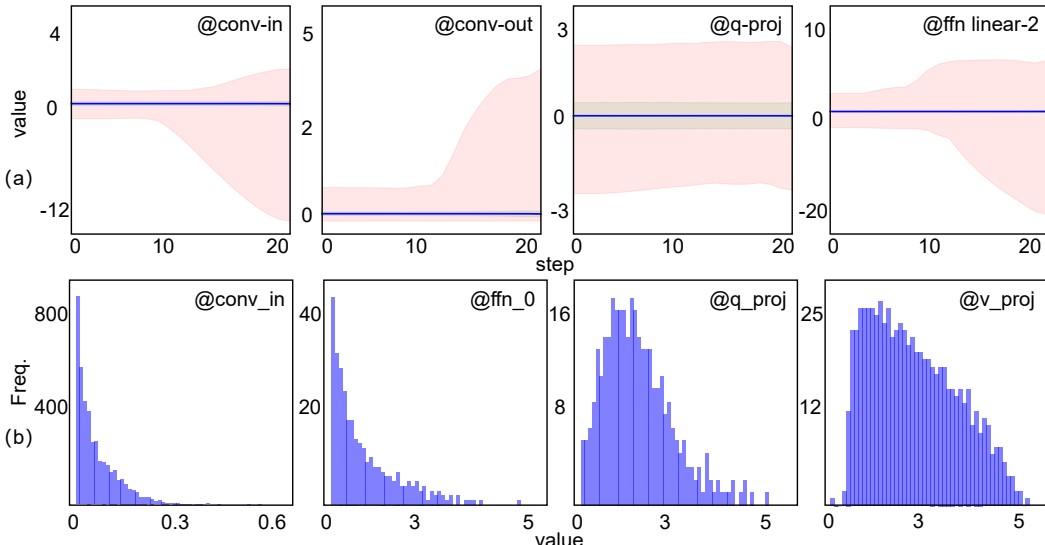

Figure 5: (a) In CMs, the input distribution in some layers varies significantly across iterative steps, while in others remains relatively stable. (b) The singular value distributions of weights differ markedly across layers: some exhibit a long-tailed pattern, while others show a uniform one.

LoRA approaches fail to account for these inherent layerwise differences in CMs, and assign the same rank to all layers. What is more concerning, this problem becomes more noticeable when quantization is incorporated: quantization errors differ greatly across different layers, which further increases the gaps between layers and thus undermines the reasonableness of treating all layers equally.

We conduct a detailed analysis of the weight distribution across different layers and how activations change with CMs iterative steps. Moreover, we examine how weights in different layers evolve throughout the training process. In summary, we have the following findings:

- The activation distributions across different layers exhibit two patterns with the iterative steps of CMs, as shown in Fig. 5 (a): the input distribution in some layers varies significantly across iterative steps, while remaining relatively stable in others.

- The singular value distributions of weights vary considerably across different layers, as shown in Fig. 5 (b): some layers exhibit a distinct long-tailed distribution, while others show a more uniform pattern.

- During training, weights in some layers converge rapidly, whereas others oscillate consistently throughout the process, as shown in Fig. 6.

Based on these findings, we propose Quantization-Aware Rank Adaptation (QRA), which assigns distinct ranks to different layers. For layers where activations exhibit minimal fluctuations across iterations and the weight singular value distribution shows a distinct long-tailed pattern, a lower rank is used to construct LoRA matrices $A$ and $B$. Conversely, for layers with significant activation fluctuations across iterations and a more uniform weight singular value distribution, we adopt a higher rank to maintain the expressive capacity of the corresponding layers and enhance robustness against quantization errors. QRA minimizes the impact of quantization on weights without increasing the total number of LoRA parameters, the detailed algorithm process is provided in Appendix A.3.

## 5 EXPERIMENT

**Baseline** This study focuses on the quantization of CMs. Therefore, we conduct a comprehensive comparison between our method and the FP16 as well as quantized versions of several competitive CMs, including LCM Luo et al. (2023a), PCM Wang et al. (2024), TCD Zheng et al. (2024a), and TDD Wang et al. (2025). Furthermore, we benchmark our method against PTQ methods, such as QDiffusion Li et al. (2023a), MixDQ Zhao et al. (2024), and SVDQuant Li et al. (2024). Additionally, we compare our method with existing LoRA-based QAT methods, including QLoRA Dettmers et al.

Table 1: Comparison of the FID, Pickscore, and IC metrics for various methods when applied to the SDXL model, utilizing the MS-COCO-10K dataset.

| Method | Steps | Bits | CFG=1 | | | CFG=2 | | | CFG=3 | | |
|---|---|---|---|---|---|---|---|---|---|---|---|
| | | | FID↓ | PickScore↑ | IC↑ | FID↓ | PickScore↑ | IC↑ | FID↓ | PickScore↑ | IC↑ |
| PCM | 4 | FP16 | 44.01 | 52.04 | 43.59 | 45.16 | 52.11 | 42.78 | 44.85 | 50.33 | 50.77 |
| TCD | 4 | FP16 | 30.98 | 70.92 | 51.82 | 31.05 | 71.92 | 50.45 | 32.09 | 69.22 | 50.31 |
| TDD | 4 | FP16 | 33.98 | 68.54 | 50.88 | 33.12 | 67.74 | 49.87 | 35.62 | 67.04 | 48.83 |
| PCM + RTN | 4 | W4A4 | 176.93 | 7.24 | 34.38 | 176.24 | 7.60 | 33.31 | 177.32 | 7.81 | 34.55 |
| TCD + RTN | 4 | W4A4 | 105.54 | 49.65 | 30.62 | 107.66 | 49.35 | 30.51 | 108.41 | 48.55 | 30.34 |
| TDD + RTN | 4 | W4A4 | 141.13 | 44.53 | 17.08 | 143.23 | 44.13 | 16.88 | 143.13 | 45.01 | 17.23 |
| QDiffusion | 20 | W4A8 | 31.09 | 35.39 | 50.96 | 31.69 | 35.19 | 50.43 | 31.19 | 34.39 | 51.26 |
| MixDQ | 20 | W4A8 | 41.88 | 49.32 | 40.92 | 40.12 | 49.18 | 39.98 | 41.55 | 48.68 | 41.23 |
| QDiffusion | 4 | W4A8 | 99.02 | 3.44 | 68.44 | 99.34 | 4.64 | 67.12 | 97.02 | 5.14 | 68.85 |
| MixDQ | 4 | W4A8 | 264.78 | 2.57 | 65.60 | 261.78 | 2.87 | 66.61 | 269.78 | 2.17 | 67.60 |
| PCM + Qalora | 4 | W4A16 | 48.22 | 50.14 | 39.23 | 49.38 | 50.84 | 37.18 | 48.91 | 47.14 | 45.44 |
| TCD + Qalora | 4 | W4A16 | 33.56 | 69.16 | 50.64 | 34.23 | 69.64 | 48.09 | 33.94 | 69.64 | 47.84 |
| TDD + Qalora | 4 | W4A16 | 36.79 | 65.66 | 50.47 | 36.71 | 65.64 | 50.34 | 37.03 | 65.64 | 46.94 |
| PCM + Qlora | 4 | W4A16 | 48.01 | 50.67 | 40.04 | 49.45 | 51.44 | 37.65 | 48.11 | 47.84 | 46.12 |
| TCD + Qlora | 4 | W4A16 | 33.76 | 69.21 | 50.33 | 34.88 | 69.14 | 47.95 | 33.44 | 69.14 | 47.85 |
| TDD + Qlora | 4 | W4A16 | 35.24 | 66.34 | 46.04 | 35.78 | 66.84 | 46.18 | 36.91 | 66.84 | 46.44 |
| Ours | 4 | W4A4 | **30.56** | **72.16** | textbf51.62 | **31.62** | **72.16** | **51.60** | **31.71** | **69.66** | **51.04** |

(2023) and QALoRA He et al. (2023). We adopt per-channel symmetric quantization for weights and per-tensor asymmetric quantization for activations. When reporting the results, we provide the metrics for both a 4-step reduced iteration and standard 20-step image generation process.

**Models and Datasets.** We select two types of models for comparative and ablation studies. First, we chose the widely used traditional Unet-based Ronneberger et al. (2015) diffusion models, SD-1.5 Rombach et al. (2022) and SDXL Podell et al. (2023). Second, we included the recently emerging and popular FLUX-1.0 Labs (2024) with a Transformer architecture. Following the TDD, we utilize the Laion-5B Schuhmann et al. (2022) High-Res dataset for the training. Additionally, we assess the performance by employing the COCO-2014 Lin et al. (2014) validation set, which is divided into two subsets with 10K captions (MS-COCO-10K) and 5K captions (MS-COCO-5K) in different experiments. To evaluate image quality, we use the Frechet Inception Distance (FID) (Jayasumana et al., 2024). To assess the richness of image content, we employ the Image Complexity Score (IC) (Feng et al., 2022). Moreover, we apply PickScore (Kirstain et al., 2023) to gauge human preference.

**Main Experimental Results.** Table 1 presents the experimental results for SDXL. We evaluate the FID, PickScore and IC of images generated by different methods. The evaluations are carried out on the MS-COCO-10k dataset under three distinct Classifier-Free Guidance (CFG) Ho & Salimans (2022) configurations. For prevalent CMs including PCM, TCD, and TDD, we evaluate the generated images under two configurations: FP16, and 4-bit quantization for both weights and activations (W4A4). It can be observed that, under the FP16 configuration, CMs typically exhibit FID metrics of approximately 40 and IC metrics around 50. Regarding the PickScore, PCM scores roughly 50, while TCD and TDD achieve scores around 70. In contrast, applying RTN quantization to these methods leads to drastic performance degradation: All FID metrics exceed 100, while both the IC and PickScore values drop significantly. For other PTQ-based quantization methods (e.g., QDiffusion and MixDQ), all metrics perform well when the number of iterative steps is 20. However, when the iterative steps are reduced to 4, all metrics deteriorate significantly.

Compared to PTQ-based methods, QAT-based approaches QALoRA and QLoRA achieve superior FID and PickScore values for images generated via 4-step iterations under W4A16. However, it is evident that both methods exhibit performance degradation in nearly all metrics compared to the original FP16 verison. In contrast, our proposed CMQuant not only avoids performance degradation but also achieves improved results across some metrics. Our method not only achieves the best FID and PickScore but also has a relatively good IC metric.

Table 2 shows the experiments on FLUX. We also compared the CMs suitable for this model under the FP16 and W4A4 configurations. It can be found that the transformer-type FLUX demonstrates

Table 2: Comparison of the FID, Pickscore, and IC metrics for various methods when applied to the FLUX, utilizing the MS-COCO-10K dataset.

| Method | Steps | Bits | CFG=1 | | | CFG=2 | | | CFG=3 | | |
|---|---|---|---|---|---|---|---|---|---|---|---|
| | | | FID ↓ | PickScore↑ | IC↑ | FID ↓ | PickScore↑ | IC↑ | FID ↓ | PickScore↑ | IC↑ |
| PCM | 4 | FP16 | 38.33 | 66.56 | 52.34 | 39.01 | 66.58 | 52.22 | 38.79 | 65.55 | 53.13 |
| TDD | 4 | FP16 | 37.19 | 68.00 | **53.22** | 38.31 | 67.88 | 53.33 | 37.71 | 66.03 | 53.04 |
| PCM + RTN | 4 | W4A4 | 39.55 | 65.84 | 50.71 | 40.82 | 65.45 | 50.94 | 40.12 | 64.92 | 52.68 |
| TDD + RTN | 4 | W4A4 | 40.45 | 66.87 | 51.05 | 41.08 | 65.49 | 51.35 | 40.39 | 65.14 | 51.21 |
| SVDQuant | 20 | W4A4 | 27.74 | 93.76 | 41.76 | 27.76 | 93.80 | 41.66 | 27.75 | 91.44 | 41.27 |
| SVDQuant | 4 | W4A4 | 89.56 | 40.14 | 28.88 | 89.80 | 39.90 | 28.35 | 89.93 | 38.80 | 29.11 |
| TDD + Qalora | 4 | W4A4 | 38.98 | 66.43 | 51.69 | 39.52 | 66.04 | 51.43 | 38.78 | 65.73 | 51.43 |
| TDD + Qlora | 4 | W4A4 | 38.33 | 66.13 | 52.29 | 40.42 | 65.01 | 52.12 | 36.88 | 65.95 | 52.54 |
| Ours | 4 | W4A4 | **32.47** | **77.78** | 56.47 | **33.18** | **77.88** | 55.69 | **32.97** | **77.64** | **56.05** |

moderate robustness to quantization. Although various metrics decline after quantization, the decrease is not as significant as that of the Unet-based SDXL model. Currently, the most popular SVDQuant method for quantizing diffusion models also fails with 4-step image generation. Its FID exceeds 90, with both the IC and PickScore declining significantly. Similarly, QAT-based methods have an adverse impact on the original CMs methods. By contrast, our method achieves the best performance in all metrics for 4-step under W4A4.

**Ablation Study. Effect of Different Modules.** To assess the impact of each module in CMQuant, we conduct ablation experiments on the MS-COCO-5k dataset. We compare the performance of LORA-based CMQuant baseline method as we gradually add three key modules. The test results in Table 3, with further details provided in Appendix 9. As demonstrated in our experiments on both SDXL and FLUX, the incorporation of each individual module contributes to the enhancement of evaluation metrics. When all three modules are employed, optimal performance is achieved across nearly all CFG configurations. Specifically, in the FLUX model, the FID metric decreases by 8.5% , the PickScore enhances by 6%, and the IC metric shows an approximate 5.7% increase.

**Semantic Alignment.** Based on the MS-COCO-5K dataset and using the SD1.5 model as the foundation, we sampled results at the 5th, 10th, 15th and 20th iterations (equally spaced within 20 FP16 iterations) for CLIP score evaluation. For comparative analysis, we also computed CLIP scores for the outputs of each of the 4 iterations of our proposed CMQuant method.

As shown in Table 5, across all iterations, the CLIP scores Hessel et al. (2021) of images generated by CMQuant are significantly higher than those of the SD-1.5 model at the corresponding iterations. This result confirms that the samples generated by CMQuant align more accurately with the semantic connotations of prompts.

Table 5: Comparison of semantic alignment across iterations teps between the FP and the CMQuant version of SD1.5. Clipscore↑ is reported.

| infer steps | 0 | 1 | 2 | 3 |
|---|---|---|---|---|
| 16-bit | 17.83 | 18.87 | 19.79 | 25.21 |
| CMQuant | 19.13 | 19.54 | 23.67 | 29.58 |

Table 3: On the SDXL and FLUX, we conduct a comparative test of the effects of applying our three proposed modules respectively, based on the original LoRA, using the MS-COCO-5K dataset.

| Model | Method | CFG=1 | | | CFG=2 | | | CFG=3 | | |
|---|---|---|---|---|---|---|---|---|---|---|
| | | FID ↓ | PickScore↑ | IC↑ | FID ↓ | PickScore↑ | IC↑ | FID ↓ | PickScore↑ | IC↑ |
| SDXL | Lora | 35.64 | 66.6 | 49.87 | 37.12 | 66.74 | 48.87 | 36.62 | 67.04 | 49.83 |
| | TDPT | 34.67 | 68.68 | 50.16 | 35.55 | 68.62 | 49.88 | 34.57 | 68.86 | 49.87 |
| | TDPT + HGS-LoRA | 34.16 | 69.16 | 50.17 | 35.26 | 69.16 | 49.86 | **33.16** | 69.16 | 50.16 |
| | TDPT + HGS-LoRA + QRA | **32.71** | **69.16** | **50.60** | **33.98** | **69.16** | **50.07** | 33.71 | 69.16 | **50.54** |
| FLUX 1.0 | Lora | 38.29 | 66.01 | 52.02 | 39.13 | 65.94 | 52.05 | 39.31 | 66.03 | 52.04 |
| | TDPT | 36.41 | 68.08 | 53.97 | 37.88 | 67.85 | 53.96 | 37.92 | 67.88 | 53.97 |
| | TDPT + HGS-LoRA | 36.25 | 69.48 | 54.45 | 37.44 | 69.43 | 54.44 | 37.41 | 69.34 | 54.45 |
| | TDPT + HGS-LoRA + QRA | **35.87** | **70.78** | **55.07** | **36.15** | **70.40** | **55.09** | **36.11** | **70.44** | **55.05** |

Table 4: Ablation study on inference iterative steps for SD-1.5 based on the MS-COCO-5K dataset.

| Method | steps=2 | | | steps=4 | | | steps=8 | | |
|---|---|---|---|---|---|---|---|---|---|
| | FID ↓ | PickScore↑ | IC↑ | FID ↓ | PickScore↑ | IC↑ | FID ↓ | PickScore↑ | IC↑ |
| TDD | 36.25 | 69.48 | 54.45 | 37.44 | 69.43 | 54.44 | 37.41 | 69.34 | 54.45 |
| CMQuant(Ours) | **35.87** | **70.78** | **55.07** | **36.15** | **70.40** | **55.09** | **36.11** | **70.44** | **55.05** |

**Consistent Advantages at Different Steps.** Our approach, like state-of-the-art consistency models (CMs), supports sample generation with varying iterative steps. To demonstrate its superiority, we evaluated images generated in steps 2, 4, and 8. As shown in Table 4, CMQuant outperforms TDD across all metrics at each step, confirming the robustness of our approach to changes in iterative steps.

**Speed and Efficiency Testing.** We evaluated hardware efficiency by testing inference speed and memory usage of the SDXL and FLUX models on a single NVIDIA A100 GPU. As shown in Table 6, the FP16 version of SDXL takes 14.15 seconds to generate an image, while FLUX requires an even longer 74.33 seconds. In contrast, our CMQuant method reduces these times significantly: SDXL generates images in just 8.27 seconds and FLUX in 21.68 seconds, representing a 2-3x speedup in image generation. In terms of memory efficiency, our 4-bit weight quantization strategy delivers substantial benefits. Compared to their FP16 counterparts, both SDXL and FLUX achieve approximately 75% memory savings when using CMQuant. Regarding computational costs, our method offers advantages in two key aspects: low-bit computation enabled by quantization and reduced iterative steps through CM distillation. Notably, compared to the original models, our method reduces computational workload by approximately 20× for both. These results demonstrate that CMQuant achieves significant improvements in inference speed and memory efficiency through low-bit quantization and CM distillation.

Table 6: Speed and Memory Testing.

| Model | SDXL | | | FLUX | | |
|---|---|---|---|---|---|---|
| | speed (img/sec) | Mem.(GB) | TFLOPs | speed (img/sec) | Mem.(GB) | TFLOPs |
| FloatPoint16 | 14.15 | 4.73 | 3.931 * 20 steps | 74.33 | 22.68 | 56.21 * 20 steps |
| Ours | 8.27 | 1.28 | 0.985 * 4 steps | 21.68 | 5.76 | 14.23 * 4 steps |
| Saving / SpeedUp | 1.71× | 27% | 3.99× * 5 | 3.43× | 25% | 3.95× *5 |

# 6 CONCLUSION

Consistency Models (CMs) accelerate diffusion models by reducing sample generation iterations, but they still have high per-iteration computational costs and large parameter sizes, which limit deployment on resource-constrained devices. Quantization, an effective model compression technique with notable success in Large Language Models, fails to work well when directly applied to CMs. This is mainly due to two issues: accumulation of quantization errors across CMs inference iterations, and incompatibility between existing LoRA and quantization in CMs.

To address these challenges, we propose CMQuant, a novel Quantization-Aware Parameter-Efficient Fine-Tuning framework for CMs with three core modules: Trajectory Distillation with Phased Targets, which splits the PF-ODE trajectory into sub-trajectories and assigns distinct optimization targets. It uses outputs from untrained quantized CMs for early stages to ensure accurate starting points and floating-point model outputs for others to align with high-precision goals, minimizing cross-iteration error accumulation. Hessian-Guided SVD-Initialized LoRA, which leverages Hessian-guided decomposition to freeze weight components that drive quantization errors, initializing LoRA with remaining components to guide updates toward quantization-friendly directions. Quantization-Aware Rank Adaptation, which adapts LoRA ranks based on layerwise variations in activations, weight distributions, and training convergence rates, minimizes the impact of quantization without increasing total LoRA parameters.

CMQuant achieves the first 4-bit quantization of CMs for both weights and activations. Experiments show that CMQuant outperforms baselines at least FID↓/PickScore↑/IC↑ of 4.74/11.61/3.01 on FLUX. Furthermore, it improves throughput by 1.71×/3.43× on SDXL/FLUX, with only 27%/25% memory footprint respectively.

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

# A APPENDIX

## A.1 LLM DISCLAIMER

The authors gratefully acknowledge the support of large language model (LLM) tools, which have assisted in refining the paper's text and optimizing its expression.

## A.2 DETAILS OF LoRA INITIALIZATION

When considering the weight quantization error, to minimize the output result error, it is necessary to optimize the following objective function:

$$\underset{W_q}{\mathrm{argmin}}\|(W_q - W)x\|^2 \tag{6}$$

Where $W$ represents the floating-point weight, $W_q$ represents the corresponding quantized weight, and $x$ is the activation of the current layer. This objective function can be equivalently transformed into:

$$\underset{W_q}{\mathrm{argmintr}}((W_q - W)xx^T(W_q - W)^T) \tag{7}$$

Where $xx^T$ is the Hessian matrix $H$ of the activation, and the formula can be further transformed into

$$\underset{W_q}{\mathrm{argmintr}}((W_q - W)H(W_q - W)^T) \tag{8}$$

Since $H$ is a positive semi-definite matrix, it has the property $H = H^{\frac{1}{2}}H^{\frac{1}{2}}, H^{\frac{1}{2}T} = H^{\frac{1}{2}}$. The equation can be further expressed as:

$$\underset{W_q}{\mathrm{argmin}}\|W_q H^{\frac{1}{2}} - wH^{\frac{1}{2}}\|^2 = \underset{W_q}{\mathrm{argmintr}}([(W_q - W)H^{\frac{1}{2}}][(W_q - W)H^{\frac{1}{2}}]^T) \tag{9}$$

Therefore, when considering the correction of quantization error, the weight we truly need to focus on is $WH^{\frac{1}{2}}$. Therefore, we perform Singular Value Decomposition (SVD) on it to obtain $W^+$, with the aim of identifying quantization-sensitive principal components and preventing them from being affected by quantization:

$$W^+ = USV = \mathrm{SVD}(WH^{\frac{1}{2}})H^{-\frac{1}{2}} \tag{10}$$

To ensure the stability in the initial stage of training, the final LoRA weight $A$ and $B$ is initialized as $W - W^+$, so as to keep the overall weight value unchanged.

## A.3 ALGORITHM OF LAYERWISE RANK ADAPTATION

To distinguish between the use of low-rank and high-rank initialization, we adopt the 90th percentile as the threshold. Additionally, we set the default value of $r$ to 64. Experimental results indicate

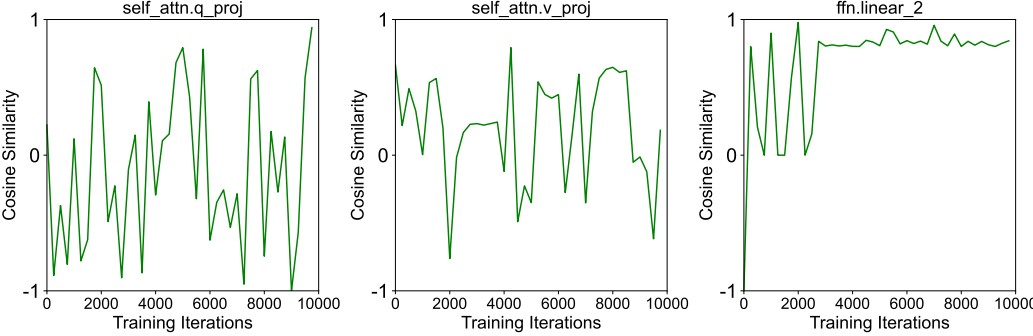

Figure 6: Curves of cosine similarity between weights at different training iterations.

that this percentile already yields favorable metrics, although more refined parameter tuning could potentially lead to even better performance.

---

**Algorithm 1** Adaptive LoRA Weight Decomposition

---

**Input:** Original weight matrix $W \in \mathbb{R}^{M \times N}$, weight set $\mathcal{W} = \{W_1, W_2, \ldots, W_m\}$, LoRA rank $r$, layer count $n$, input prompts $\mathcal{P}$, geneate imgs need iters $S$
**Output:** LoRA matrices $A$, $B$, frozen weight matrix $W_{\text{frozen}}$

    **Activation Collection Module**
1: Initialize Hessian list $\mathcal{H} \leftarrow \{\}$, input list $\mathcal{I}_{\text{inp}} \leftarrow \{\text{layer\_id} : [] \, for\_in S]\}$
2: **for** $p \in \mathcal{P}$ **do**
3:     out $\leftarrow$ model$(p)$
4:     Record layer inputs in $\mathcal{I}_{\text{inp}}$
5: **end for**
    **Variance Analysis Module**
6: Compute activation variance across time steps:
7: **for** $l \in$ layers **do**
8:     $v_l \leftarrow$ cal\_var$(\mathcal{I}_{\text{inp}}[l])$
9: **end for**
10: Sort variances: $\mathcal{V}_{\text{sorted}} \leftarrow$ sorted$(v_l, \text{descending=False})$
11: High-variance layers: $\mathcal{L}_{\text{high}} \leftarrow \{l \mid v_l > \mathcal{V}_{\text{sorted}}[0.9|\mathcal{V}_{\text{sorted}}|]\}$
12: Low-variance layers: $\mathcal{L}_{\text{low}} \leftarrow \{l \mid v_l < \mathcal{V}_{\text{sorted}}[0.1|\mathcal{V}_{\text{sorted}}|]\}$
    **Hessian Calculation Module**
13: **for** idx, $l \in$ layers **do**
14:     Compute Hessian: $H_l \leftarrow \text{inp}[l] \times \mathcal{I}_{\text{inp}}[l]^T$
15:     Update: $\mathcal{H}[l] \leftarrow H_l + \mathcal{H}[l] \times \frac{\text{idx}}{n}$
16: **end for**
    **Weight Analysis Module**
17: Compute weight rank variance:
18: **for** $l, W_l \in \mathcal{W}$ **do**
19:     $R, V \leftarrow$ eig$(W_l)$
20:     $\sigma_l \leftarrow$ cal\_var$(R)$
21: **end for**
22: Sort variances: $\mathcal{S}_{\text{sorted}} \leftarrow$ sorted$(\sigma_l, \text{descending=False})$
23: High-variance layers: $\mathcal{W}_{\text{high}} \leftarrow \{l \mid \sigma_l > \mathcal{S}_{\text{sorted}}[0.9|\mathcal{S}_{\text{sorted}}|]\}$
24: Low-variance layers: $\mathcal{W}_{\text{low}} \leftarrow \{l \mid \sigma_l < \mathcal{S}_{\text{sorted}}[0.1|\mathcal{S}_{\text{sorted}}|]\}$
    **LoRA Initialization Module**
25: **for** $l, W_l \in \mathcal{W}$ **do**
26:     **if** $l \in \mathcal{W}_{\text{high}} \cup \mathcal{L}_{\text{high}}$ **then**
27:         $U, S, V \leftarrow$ svd\_lowrank$(W_l \times \sqrt{\mathcal{H}[l]}, 1.5r)$
28:     **else if** $l \in \mathcal{W}_{\text{low}} \cup \mathcal{L}_{\text{low}}$ **then**
29:         $U, S, V \leftarrow$ svd\_lowrank$(W_l \times \sqrt{\mathcal{H}[l]}, 0.5r)$
30:     **else**
31:         $U, S, V \leftarrow$ svd\_lowrank$(W_l \times \sqrt{\mathcal{H}[l]}, r)$
32:     **end if**
33:     Initialize LoRA: $W_{\text{frozen}} = USV \times \frac{1}{\sqrt{\mathcal{H}[l]}}, \quad AB = W - W_{\text{frozen}}$
34: **end for**
35: **return** $A$, $B$, $W_{\text{frozen}}$

---

### A.3 SUPPLEMENTARY EXPLANATION: LOGICAL CONNECTION BETWEEN SINGLE-STEP QUANTIZATION ERROR AND DTD TRAJECTORY PARTITIONING

Results in [7] have verified that when any single step of the FLUX-PCM model is individually quantized, the fluctuation of FID is no more than 0.74 (e.g., FID = 39.75 when quantizing step = 0, which is only 0.74 higher than that of the floating-point model) and the drop in PickScore is no more than 0.64. This indicates that the absolute error introduced by quantization within a single step is inherently within the tolerable range of the model. However, the iterative inference characteristic of

Table 7: Compare the quantization effect of a specific iteration of LCM for FLUX, based on the MS-COCO-5K dataset under the configuration of CFG=2.

| | Floating-Point | Quant$_{step}=0$ | Quant$_{step}=1$ | Quant$_{step}=2$ | Quant$_{step}=3$ |
|---|---|---|---|---|---|
| FID ↓ | 39.01 | 39.75 | 39.64 | 39.23 | 39.34 |
| PickScore↑ | 66.58 | 65.94 | 66.28 | 66.34 | 66.44 |
| IC↑ | 52.22 | 51.64 | 51.73 | 51.95 | 52.09 |

time t serves as input at time $t-1$, and inaccuracies in the input at $t-1$, interacting with quantization errors, introduce further inaccuracies into the output at $t-1$, with error accumulation ultimately rooted in output inaccuracies at later times. As shown in Table 1, methods like PCM + RTN, TCD + RTN, and TDD + RTN, suffer severe performance degradation—FID values skyrocket (e.g., PCM + RTN reaches 176.93 under CFG=1) and PickScore/IC plummet (e.g., PCM + RTN's PickScore is only 7.24 under CFG=1). This demonstrates that when the floating-point model's output is naively used as the fitting target for earlier steps (forcing the quantized model to learn the mapping from floating-point trajectory to quantized output), the inability of low-bit spaces to fully cover floating-point trajectory distributions leads to tiny early-step errors being magnified through subsequent iterations, devastating overall performance.

The trajectory partitioning logic of Trajectory Distillation with Phased Targets (TDPT) is explicitly designed around a core insight: single-step errors are controllable, but multi-step error spread causes significant damage. For early sub-trajectories, using the quantized model's output as the fitting target allows the model to form a self-consistent local trajectory within the low-bit space. At this stage, the model does not need to adapt to the high-dimensional distribution of the floating-point trajectory; it only needs to ensure output consistency between adjacent steps in the quantized space, thus eliminating errors caused by forcing the quantized model to fit the floating-point trajectory. For late sub-trajectories, we switch to using the floating-point model's output as the target because the model is near convergence by this point, with a more stable output distribution. The quantized space can adequately capture the local features of the floating-point target, and errors in late steps have no subsequent iterations to propagate through, preventing significant accumulation. This design—adapting to the quantized space in early stages and aligning with floating-point goals in later stages—capitalizes on the characteristic that single-step errors are manageable while multi-step spread is harmful, fundamentally blocking long-range accumulation of quantization errors.

In our experiments, since image generation is based on 4-step iteration, we set the target of the final step as the output of the floating-point model, while using quantized model outputs for other steps. This corresponds to $T'=1$ in Equation 4. For image generation tasks with different total iterative steps, fine-tuning the value of $T'$ may yield better results. Based on empirical observations from our quantization experiments across multiple models, we recommend setting $T'$ to approximately 1/4 of the total iterative steps, which balances the need to suppress error propagation in early stages and align with high-precision targets in late stages.

## A.5 DISCUSSION ON THE EFFECTIVENESS OF THE IC

As shown in Fig. 7, the first row presents images generated by different methods, while the second row displays the corresponding Image Complexity (IC) analysis results for these images. In the IC analysis images, bright-colored regions indicate high complexity, whereas purple regions denote low complexity. It can be observed that images generated by the TCD under the W4A4 quantization setting contain numerous noise points and exhibit unclear content. Despite this, the model achieves an IC value of 35. Both the central object regions and the noisy areas throughout the image show high complexity. By contrast, images generated by our method not only align closely with semantic descriptions but also feature sharp clarity. They also have a clean background free from noise interference and outperform the TCD model in both CLIP scores and Frechet Inception Distance (FID) metrics. However, our method yields a lower IC value. As discussed in the experimental section, we find that images with high IC values do not always correspond to high visual quality. The IC metric only holds certain reference value when the FID, CLIP score, and Pickle score of compared images are comparable.

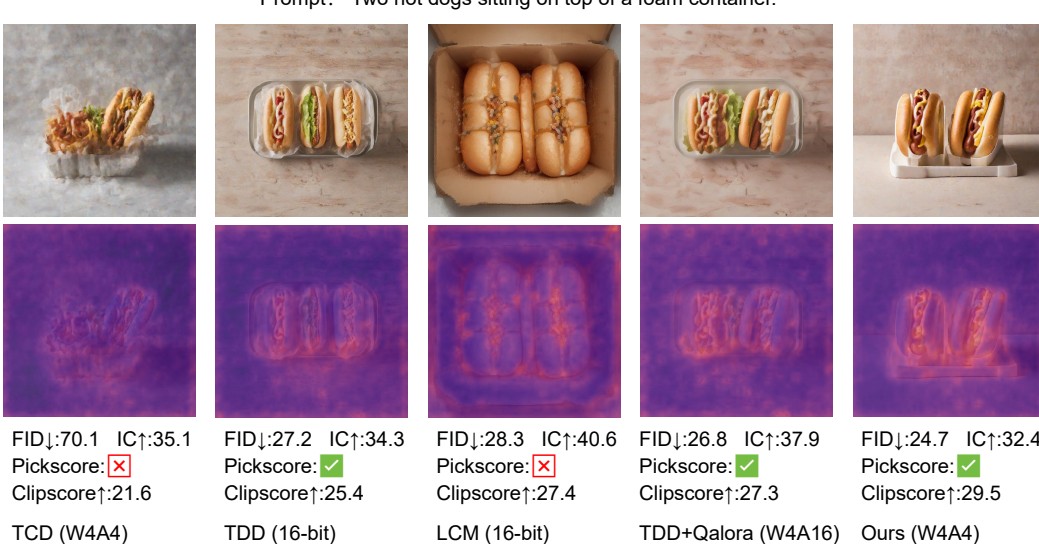

Figure 7: Images generated by different methods on SD1.5 models and the corresponding complexity analysis images.

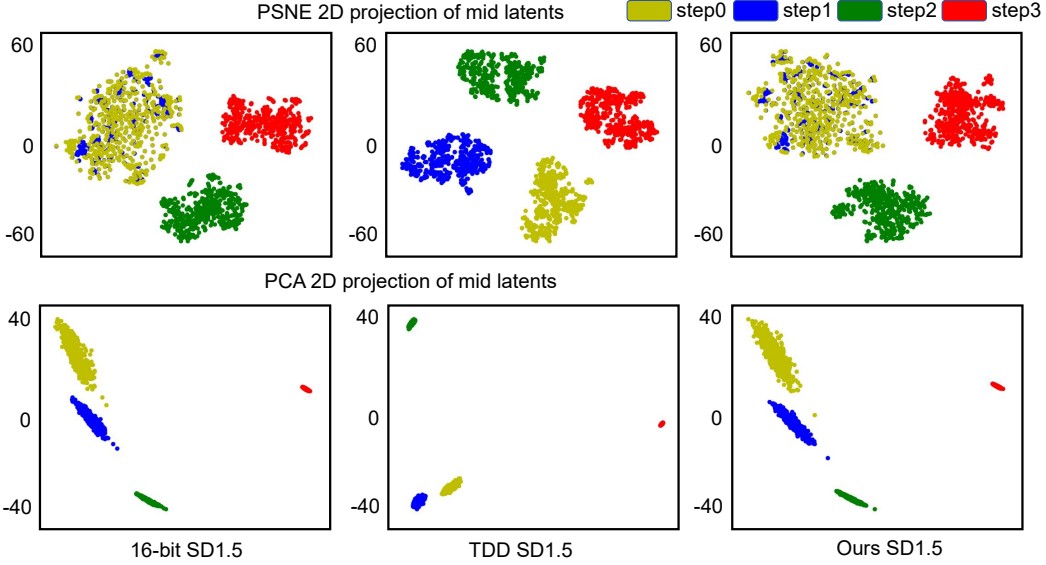

Figure 8: Distribution diagrams of the reduced-dimensional intermediate latent outputs of SD1.5 at different times under different methods

### A.6 INTERMEDIATE FEATURE DISTRIBUTION

As demonstrated in the ablation experiments of the experimental section, our proposed CMQuant outperforms the floating-point model by achieving higher CLIP scores at all iterations. Under identical experimental configurations, we employed two techniques—Principal Component Analysis (PCA) and Probabilistic Nearest Neighbor Embedding (PNSE)—to conduct dimensionality reduction and visualization on the intermediate features of the generated images. As shown in Fig. 8, regardless of whether PCA or PNSE is adopted for dimensionality reduction, the intermediate feature distribution of the CMQuant method closely mirrors that of the original floating-point model. In contrast, for the TDD method, while the difference in the final number of iterative steps from the floating-point model is not particularly significant, its feature distribution diverges substantially from the latter during intermediate iterative steps.

## A.7 EFFECTIVENESS ON LARGE LANGUAGE MODEL

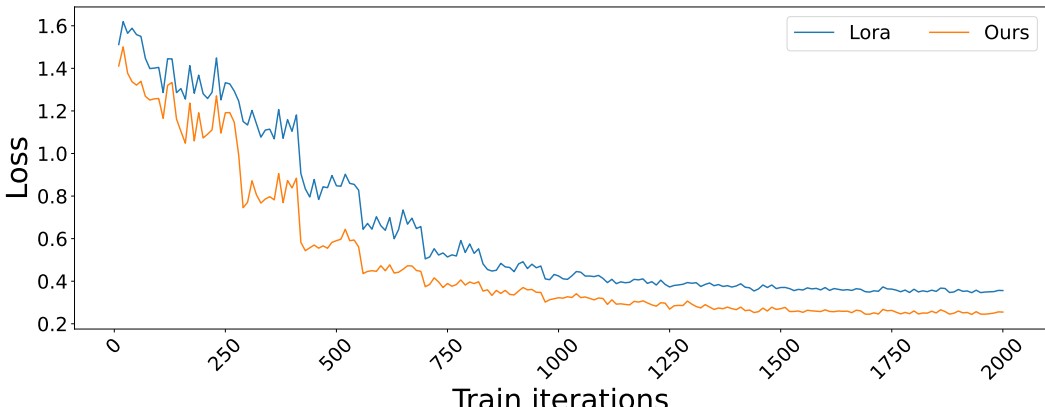

Figure 9: The loss change curves of fine-tuning the LLAMA3-8b model using the default LoRA method and our method on the Alpaca dataset.

Inspired by LoRA-based PEFT widely used in large language model fine-tuning, we applied our CMQuant method without TDPT to optimize the llama3-8B model on the Alpaca dataset (Taori et al., 2023) via LLaMA-Factory's (Zheng et al., 2024b) codebase. Using 2,000 training steps for both default LoRA and CMQuant, Fig. 9 shows that CMQuant achieves a faster loss decline and a lower final convergence loss than LoRA. This result is highly valuable, demonstrating CMQuant's applicability to large language model fine-tuning. In future research, we will further explore and deepen the application of the CMQuant in the fine-tuning of large language models.

## A.8 DERIVATION OF THE EFFECTIVENESS OF TDPT

Let the ideal trajectory be $\{x_k\}_{k=0}^N$, where $x_k = f_\theta(x_{k+1})$ and $x_0$ is the final generated sample. The quantized model produces a perturbed trajectory $\{\tilde{x}_k\}_{k=0}^N$, where $\delta(\cdot)$ is the quantization perturbation function, assumed to be bounded: $\|\delta(x)\| \leq \epsilon$ for all $x$.

$$\tilde{x}_k = \tilde{f}(\tilde{x}_{k+1}) = f_\theta(\tilde{x}_{k+1}) + \delta(\tilde{x}_{k+1}).$$

The cumulative output error at step $k$ is defined as:

$$e_k = \tilde{x}_k - x_k \tag{11}$$
$$= [f_\theta(\tilde{x}_{k+1}) + \delta(\tilde{x}_{k+1})] - f_\theta(x_{k+1})$$
$$= \underbrace{[f_\theta(\tilde{x}_{k+1}) - f_\theta(x_{k+1})]}_{\text{Error due to deviated input}} + \underbrace{\delta(\tilde{x}_{k+1})}_{\text{Quantization perturbation}} \tag{12}$$

Assuming $f_\theta$ is Lipschitz continuous with constant $L > 0$, we have:

$$\|f_\theta(\tilde{x}_{k+1}) - f_\theta(x_{k+1})\| \leq L\|\tilde{x}_{k+1} - x_{k+1}\| = L\|e_{k+1}\|. \tag{13}$$

Combining 12 and 13, the error norm is bounded by the following first-order difference inequality:

$$\|e_k\| \leq L\|e_{k+1}\| + \epsilon, \quad \text{for} \quad k = 0, 1, \ldots, N-1, \tag{14}$$

with the boundary condition $\|e_N\| = 0$, since the input content is identical.

Solving the difference inequality 14 by backward recursion from $k = N-1$ to $k = 0$ yields:

$$\|e_{N-1}\| \leq \epsilon$$
$$\|e_{N-2}\| \leq L\epsilon + \epsilon = \epsilon(1+L)$$
$$\|e_{N-3}\| \leq L\epsilon(1+L) + \epsilon = \epsilon(1+L+L^2)$$
$$\vdots$$
$$\|e_0\| \leq \epsilon \sum_{i=0}^{N-1} L^i. \tag{15}$$

For $L \neq 1$, this geometric series has a closed-form solution:

$$\|e_0^{\text{QAT}}\| \leq \epsilon \cdot \frac{L^N - 1}{L - 1}. \tag{16}$$

This result mathematically confirms the paper's observation: even if the single-step error $\epsilon$ is small (as shown in Table 7), the final error $\|e_0^{\text{QAT}}\|$ grows exponentially with the number of steps $N$ if $L > 1$. TDPT partitions the trajectory into two segments at step $M$ (corresponding to time $T'$ in Eq. 4 of the paper). The key innovation is the change of optimization target in the early phase ($k \geq M$). In the early phase, TDPT optimizes the model to satisfy:

$$f_\theta(x_t, t) \approx f_{Q(\theta)}(x_{t'}, t'), \quad \forall t, t' \in [T', T], \tag{17}$$

where $f_{Q(\theta)}$ is the *untrained* quantized model. In our discrete model, this means for $k \geq M$:

$$f_\theta(\tilde{x}_{k+1}) \approx f_{Q(\theta)}(\tilde{x}_{k+1}) = f_\theta(\tilde{x}_{k+1}) + \delta_{\text{init}}(\tilde{x}_{k+1}), \tag{18}$$

where $\delta_{\text{init}}$ is the initial, untrained quantization perturbation. After training, the model learns a compensatory mapping such that:

$$f_\theta(\tilde{x}_{k+1}) + \delta_{\text{init}}(\tilde{x}_{k+1}) \approx f_\theta(x_{k+1}). \tag{19}$$

Substituting 19 into the error definition 12 for $k \geq M$:

$$e_k = [f_\theta(\tilde{x}_{k+1}) + \delta_{\text{init}}(\tilde{x}_{k+1})] - f_\theta(x_{k+1}) \approx 0.$$

Therefore, TDPT effectively *resets* the error at the partition point $k = M$:

$$\|e_M^{\text{TDPT}}\| \approx 0. \tag{20}$$

This is the core mathematical operation of TDPT: it constructs a local, self-consistent manifold in the quantized space, preventing the initial propagation of cross-space fitting errors.

In the late phase, TDPT reverts to the standard objective of aligning with the floating-point model. The error dynamics for $k < M$ revert to the standard form in 14:

$$\|e_k\| \leq L\|e_{k+1}\| + \epsilon, \quad \text{for} \quad k = 0, 1, \ldots, M - 1, \tag{21}$$

but now with the new, TDPT-induced boundary condition from 20: $\|e_M\| \approx 0$. Solving the difference inequality 21 from $k = M - 1$ to $k = 0$ with $\|e_M\| \approx 0$:

$$\|e_0^{\text{TDPT}}\| \leq \epsilon \sum_{i=0}^{M-1} L^i = \epsilon \cdot \frac{L^M - 1}{L - 1} \quad (\text{for } L \neq 1). \tag{22}$$

Comparing the final error bounds of Standard QAT 16 and TDPT 22:

$$\frac{\|e_0^{\text{TDPT}}\|}{\|e_0^{\text{QAT}}\|} \leq \frac{L^M - 1}{L^N - 1}. \tag{23}$$

## A.9 LIMITATIONS

Table 8 compares FID, Pickscore, and IC metrics of LCM, PCM, TDD, and our method on MS-COCO-10K for SD1.5. Our method outperforms in FID and IC across CFG conditions, while LCM leads in Pickscore. We hypothesize that this observation is associated with the dataset and training duration employed by the LoRA model in the latest version of LCM. Additionally, the SD-1.5's Unet-based convolutional structure with a relatively small number of parameters is highly sensitive to quantization, potentially contributing to this outcome. We will further investigate the reasons and improve CMQuant to boost its Pickscore on SD1.5.

## A.10 OVERALL ABLATION STUDY

This table presents an ablation study investigating the effects of three components (TDPT, HGS-LoRA, and QRA) on two models (SDXL and FLUX 1.0) under CFG = 1. Performance is measured via FID (lower is better), PickScore (higher is better), and IC (higher is better), with checkmarks denoting enabled components. It can be seen that the combination of any two methods achieves better performance than a single method.

Table 8: Comparison of the FID, Pickscore, and IC metrics for various methods when applied to the SD1.5, utilizing the MS-COCO-10K dataset.

| Method | Steps | Bits | CFG=1 | | | CFG=2 | | | CFG=3 | | |
|--------|-------|------|-------|-------|-----|-------|-------|-----|-------|-------|-----|
| | | | FID ↓ | PickScore↑ | IC↑ | FID ↓ | PickScore↑ | IC↑ | FID ↓ | PickScore↑ | IC↑ |
| LCM | 4 | FP16 | 31.294 | **53.56** | 42.5 | 32.74 | **53.37** | 42.44 | 34.56 | **41.88** | 42.33 |
| PCM | 4 | FP16 | 28.15 | 50.40 | 48.58 | 29.15 | 50.40 | **48.58** | 29.14 | 36.48 | 40.11 |
| TDD | 4 | FP16 | 26.27 | 49.65 | 50.95 | 28.01 | 48.58 | 46.33 | 28.97 | 33.25 | 43.33 |
| Ours | 4 | W4A4 | **25.93** | 52.36 | **51.54** | **26.54** | 52.65 | 46.80 | **28.32** | 37.02 | **44.25** |

## A.11 VISUAL QUALITY DISPLAY

Figs. 10 and 11 respectively present comparisons of images generated via 4-step inference across different methods. The former contrasts images generated by our method and PCM (with SD1.5 as the base model), while the latter compares those generated by our method and TDD (using SDXL as the base model)—all under identical input prompts and random seeds. Evidently, our method outperforms both PCM and TDD in terms of image quality. When compared with PCM, our method produces images where characters have normal facial features, bicycle structures are complete, and the overall content is rich with diverse coloration. When contrasted with TDD, our method generates images with objects that have intact structures and characters with abundant fine details. Fig. 12 illustrates the ablation study results for the three modules of our CMQuant method. The results demonstrate that integrating all three modules enhances the quality of 4-step generated images, eliminating facial distortion, structural defects in objects, and logically inconsistent elements. Fig. 13 showcases the application of our CMQuant method to FLUX for 4-step image generation, which yields excellent performance across portrait, landscape, and various stylized image categories.

Table 9: On the SDXL and FLUX, we conduct a comparative test of the effects of applying our three proposed modules respectively, based on the original LoRA, using the MS-COCO-5K dataset.

| Model | TDPT | HGS-LoRA | QRA | CFG=1 | | |
|-------|------|----------|-----|-------|------------|-----|
| | | | | FID ↓ | PickScore↑ | IC↑ |
| SDXL | ✓ | | | 34.67 | 68.68 | 50.16 |
| | | ✓ | | 35.17 | 67.71 | 49.88 |
| | | | ✓ | 34.88 | 66.81 | 49.77 |
| | ✓ | ✓ | | 34.16 | 69.16 | 50.17 |
| | ✓ | | ✓ | 34.31 | 69.02 | 50.09 |
| | ✓ | ✓ | | 33.41 | 69.08 | 50.21 |
| FLUX 1.0 | ✓ | | | 36.41 | 68.08 | 53.97 |
| | | ✓ | | 36.57 | 68.29 | 53.56 |
| | | | ✓ | 36.12 | 68.42 | 53.66 |
| | ✓ | ✓ | | 36.25 | 69.48 | 54.45 |
| | ✓ | | ✓ | 36.21 | 69.34 | 54.67 |
| | ✓ | ✓ | | 36.04 | 70.02 | 54.82 |

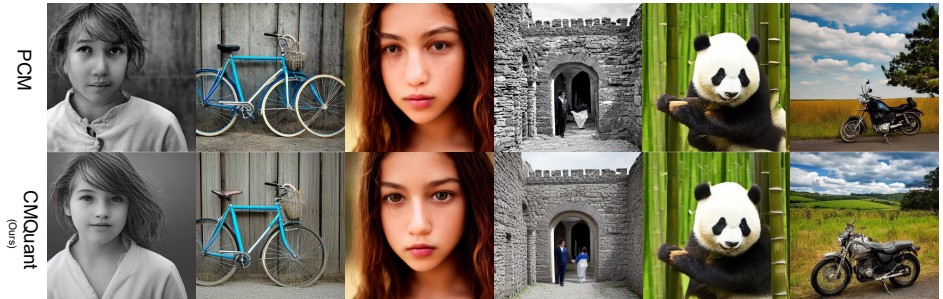

Figure 10: Visual effect comparison between our method and PCM on SD-1.5 with 4-step.

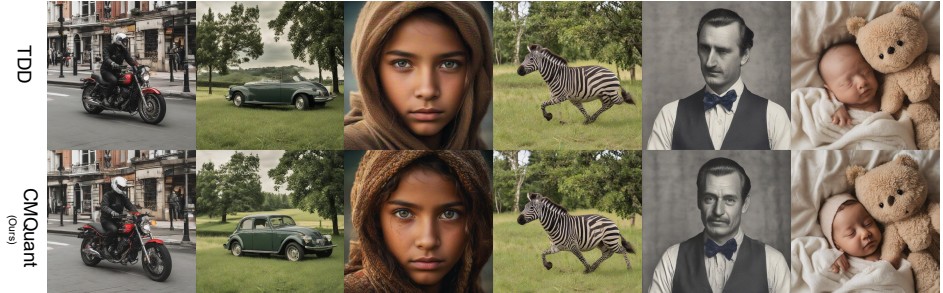

Figure 11: Visual effect comparison between our method and TDD on SDXL with 4-step.

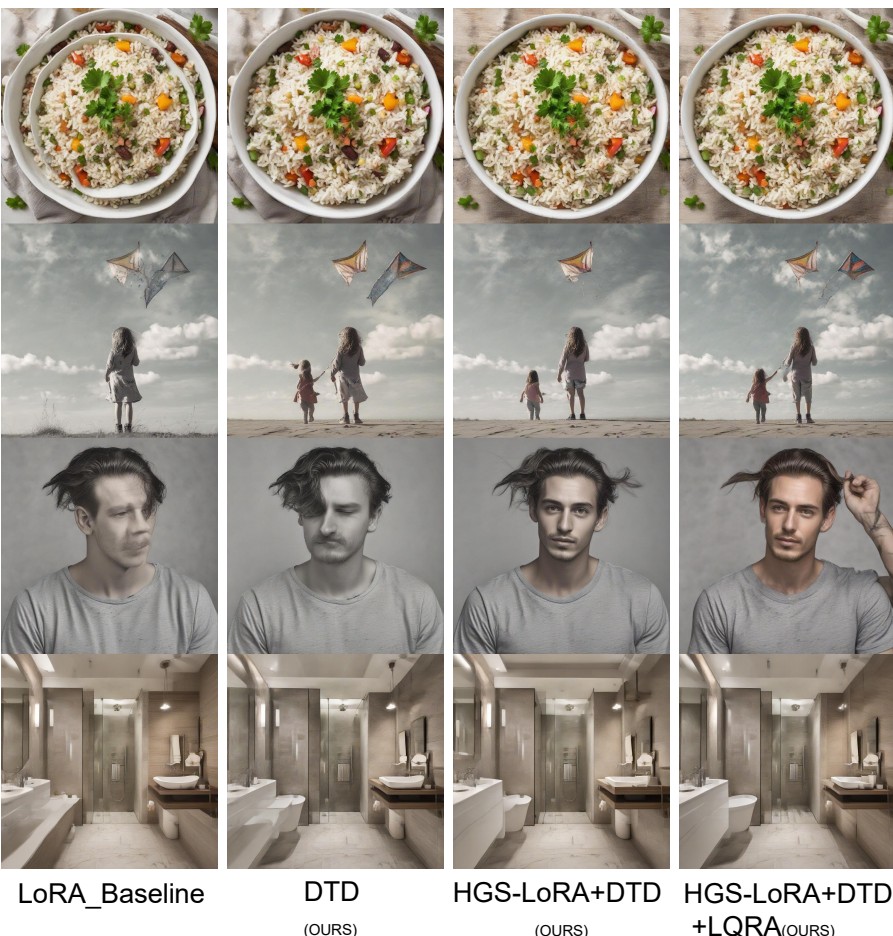

Figure 12: Visual effect comparison of the ablation experiments of our three modules on SDXL with 4-step.

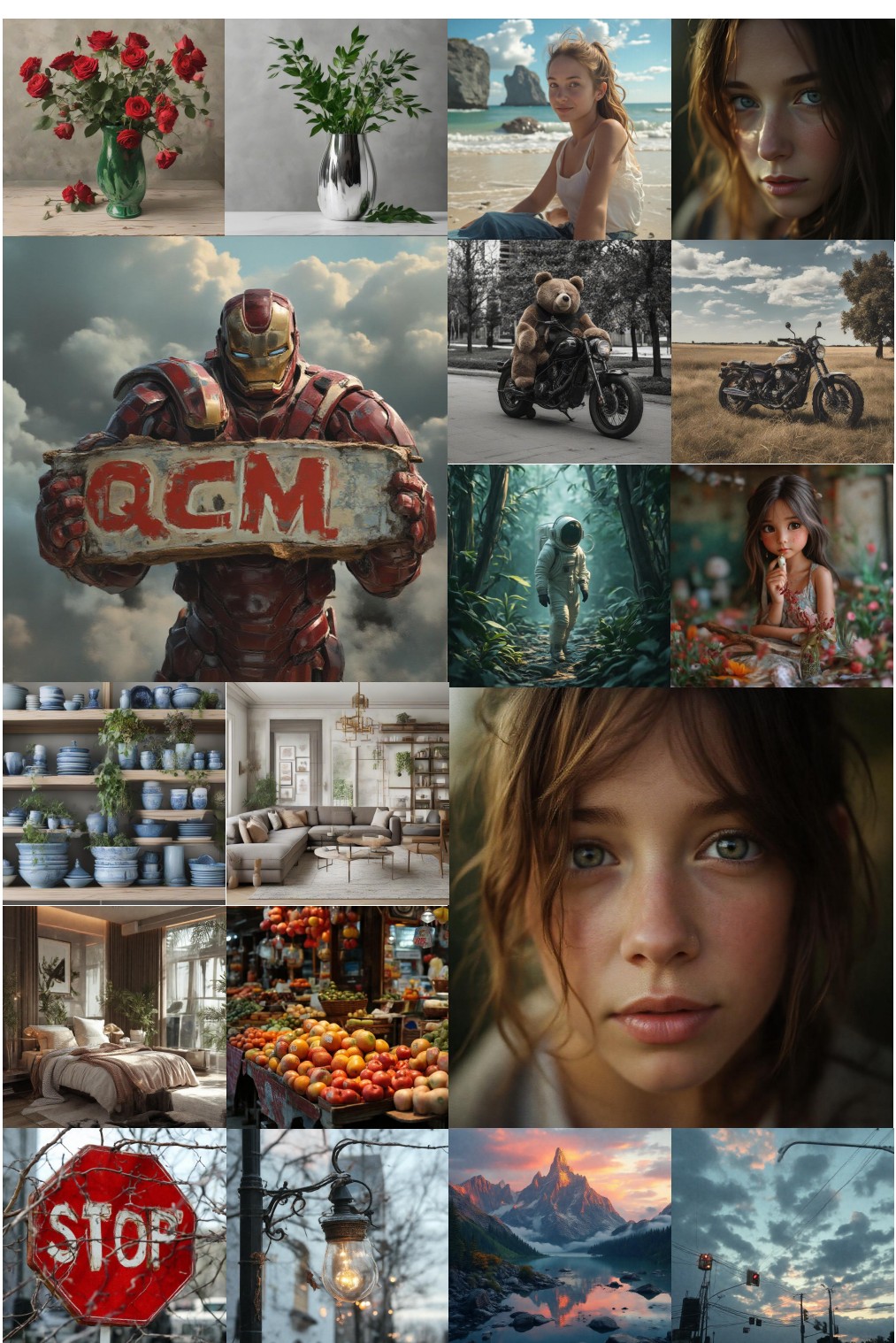

Figure 13: Visual display of the images generated by our method on the FLUX with 4-step.

