# OpenReview forum: "CMQuant: A Quantization-Aware Parameter-Efficient Fine-Tuning Framework for 4-Bit Consistency Models"
_ICLR.cc/2026/Conference — Submitted to ICLR 2026_

### Official Review · Reviewer_ECwp · 2025-10-30

**Soundness:** 3
**Presentation:** 3
**Contribution:** 2
**Rating:** 4
**Confidence:** 4

**Summary:**

This paper proposes CMQuant, a quantized parameter efficient fine-tuning method for consistency models.
CMQuant consists of three parts: (1) two-step fine-tuning with different objective; (2) activation-induced low-rank approximation to compensate weight quantization error (3) adaptive rank allocation for adapters.

**Strengths:**

The motivations, problem statement, and method are clear. For example, the observation on weight & activation statistics justifies of need of adaptive rank allocation.

**Weaknesses:**

- The contribution of the paper is unclear, especially considering recent related works not mentioned in the paper
  - The second contribution of the work, i.e., HGS-LoRA, which is an analytical activation-induced low-rank approximation, has been proposed in other works, for example, [QERA (ICML2025)](https://openreview.net/forum?id=LB5cKhgOTu)  has exactly the same solution, and [Caldera (NeruIPS2024)](https://proceedings.neurips.cc/paper_files/paper/2024/hash/a20e8451ffb07ad25282c21945ad4f19-Abstract-Conference.html) has a similar case where activation error is additionally considered.
  - The last contribution of the work, which relocates lora ranks across layers, has also been explored in other works, e.g., [AdaLoRA (ICLR2023)](https://arxiv.org/abs/2303.10512), [LQ-LoRA (ICLR2024)](https://openreview.net/pdf?id=xw29VvOMmU), etc.
- Unclear experiment design.
  - Table 1 and table 2 seems mixing the comparison of fine-tuning-based methods with post-training quantization methods? For example, in table2, SVDQuant is post-training quantization method, but CMQuant is fine-tuning based.

**Questions:**

Please answer the following questions in addition to the Weakness section
- Does CMQuant use any techniques/tricks to mitigate quantization error in activations before the fine-tuning starts? My understand is that HGS-LoRA is an weight initialization method that compensate the quantization error of weights.
- What's the number format used in this work? INT4?
- QLoRA seems a bit outdated as the baseline in Table 1 and Table 2. Could the author compare against newer quantized parameter-efficient fine-tuning methods?
- The claim of "most existing methods focus solely on weight quantization while neglecting activation quantization, leaving the model with substantial computational overhead" may be inaccurate. My understanding is that activation quantization for Transformer is also an activate research area.

---

> ### Author Response · Authors · 2025-11-19
> **Rebuttal by Authors [1/3]**
>
> Dear Reviewer,
>
> We sincerely thank you for your valuable time and efforts in reviewing our manuscript. We have addressed each comment and made the necessary revisions to improve the quality and clarity of our manuscript.
>
> **1. Concerns in the contributions**
>
> We fully acknowledge the importance of contextualizing our contributions against state-of-the-art approaches, and we appreciate the opportunity to clarify the distinctiveness of CMQuant—both in its individual modules and integrated framework—by addressing the differences from the methods you mentioned (QERA, CALDERA, AdaLoRA, LQ-LoRA).
>
> **1.1  three core modules of CMQuant**
>
> The three core modules of CMQuant (TDPT, HGS-LoRA, and QRA) are not simply concatenated; instead, they form a progressive collaborative architecture centered on "resolving the core contradictions of 4-bit quantization for Consistency Models (CMs)."
>
> Unlike existing methods designed for Large Language Models (LLMs) or vanilla diffusion models, CMQuant directly addresses the fundamental contradictions of CMs:
>
> - **Iterative trajectory fitting**: CMs generate samples via multi-step trajectory optimization, where quantization errors accumulate exponentially across iterations (Appendix A.8).
>
> - **LoRA dependency**: CMs rely heavily on LoRA for training (injecting low-rank matrices without modifying pretrained weights), making quantization prone to breaking initialization consistency and degrading convergence.
>
> - **Layer-wise dynamic heterogeneity**: Quantization amplifies differences in activation fluctuations, weight distributions, and convergence speeds across CMs layers (Figures 5-6), rendering uniform optimization ineffective.
>
> **Across iterations**: TDPT blocks error accumulation, providing a stable optimization foundation for HGS-LoRA and QRA.
>
> **Within a single iteration**: HGS-LoRA and QRA operate from the perspectives of initialization and resource allocation, respectively. HGS-LoRA guides optimization directions, while QRA allocates resources to critical layers, amplifying their synergistic effect.
>
> We detail the differences below:
>
> **1.2 HGS-LoRA vs. QERA  / CALDERA**
>
> We agree that QERA and CALDERA are excellent works in quantized low-rank adaptation, but their core goals, mathematical logic, and adaptation scenarios are fundamentally different from HGS-LoRA:
>
> **Distinct design goals:** QERA and CALDERA are two excellent works that we have thoroughly studied. Tailored for Large Language Models (LLMs), they address the core contradiction of "weight/output error loss" after LLM quantization. Leveraging activation statistics—QERA uses the autocorrelation matrix, while CALDERA employs a Hessian matrix derived from calibration data—they model output errors and act as **"error compensators"**: QERA solves for low-rank terms to minimize output errors, and CALDERA uses low-rank (LR) components to offset quantization errors of the quantized backbone (Q). Notably, both methods merely **use LoRA as a carrier for low-rank terms to compensate for quantization errors**.
>
> While HGS-LoRA also relies on an initialization-based approach, its underlying logic and core objectives are fundamentally different from those methods. It leverages the Hessian matrix of activations to measure the contribution of each weight component to quantization errors, accurately identifying and freezing the "quantization-sensitive principal components." Meanwhile, it initializes LoRA weights using the "quantization-insensitive non-principal components," guiding the model to update along "quantization-friendly directions." Without HGS-LoRA, conventional LoRA initialization would cause the distribution of quantized weights to deviate from the pretrained state.
>
> In contrast to the aforementioned methods that act as "error compensators" by fitting existing quantization errors, HGS-LoRA is more focused on reducing quantization errors at the source. Serving as **a "quantization-friendly update guide,"** it freezes sensitive principal components and initializes LoRA exclusively with non-sensitive components, ensuring that model updates do not amplify quantization errors. By deeply integrating LoRA with quantization logic and leveraging SVD decomposition, HGS-LoRA guarantees the "quantization consistency" between LoRA initialization and pretrained weights, effectively addressing the training convergence challenge of Consistency Models (CMs).

---

> > ### Author Response · Authors · 2025-11-19
> > **[2/3]**
> >
> > **1.3 QRA vs. AdaLoRA  / LQ-LoRA**
> >
> > We clarify that QRA is not a repetition of existing rank allocation methods (e.g., AdaLoRA, LQ-LoRA), but a quantization-aware rank adaptation strategy tailored for Consistency Models (CMs), with fundamental differences in design goals, decision mechanisms, and **adaptability to CMs' unique properties**:  Unlike AdaLoRA (gradient cosine similarity) or LQ-LoRA (full-precision singular values), QRA’s rank allocation is based on two quantization-specific metrics:
> >
> >   * **Activation iterative variance**: Layers with high activation fluctuations during CMs’ trajectory fitting accumulate quantization errors across iterations, requiring higher ranks;
> >   * Weight singular value distribution: Layers with uniform singular values (more vulnerable to quantization) need higher ranks than those with long-tailed distributions (robust to quantization).
> >
> > CMs generate samples via iterative trajectory fitting, where layer errors amplify across steps. Existing methods (designed for single-step inference) ignore this. QRA explicitly suppresses iterative error accumulation by prioritizing ranks for high-fluctuation layers, synergizing with TDPT to block error propagation.
> >
> > To further address the reviewer’s concerns, we conducted comparative experiments between QRA, AdaLoRA, and LQ-LoRA. Specifically, we evaluated the Frechet Inception Distance (FID, lower is better) on the SDXL model using the MS-COCO-10K dataset under the configuration of Classifier-Free Guidance (CFG) = 1. The results demonstrate that both AdaLoRA and LQ-LoRA yield valid performance improvements, while our QRA method achieves even more superior results.
> >
> > | Method                          | FID↓  |
> > |---------------------------------|-------|
> > | TDPT + HGS-LoRA                 | 34.16 |
> > | TDPT + HGS-LoRA + QRA           | 32.71 |
> > | TDPT + HGS-LoRA + AdaLoRA       | 33.17 |
> > | TDPT + HGS-LoRA + LQ-LoRA       | 33.14 |
> >
> > **1.4 TDPT**
> >
> > It alleviates the "error accumulation and amplification" issue in CMs quantization across iterations by proposing a "staged target assignment" strategy: early sub-trajectories adopt outputs from untrained quantized models as targets (ensuring self-consistency within the quantized space and avoiding cross-space fitting errors), while late sub-trajectories use outputs from floating-point models as targets (to align with high-precision generation objectives). This marks the first quantization-friendly distillation method tailored to the iterative inference characteristics of CMs, blocking error accumulation at its source.
> >
> > Without TDPT, quantization errors would accumulate exponentially during CMs iterations. Even if HGS-LoRA optimizes LoRA initialization, training would diverge due to excessive input errors. Through its staged target assignment, TDPT confines errors in early sub-trajectories within the quantized space, thereby reducing HGS-LoRA’s optimization burden to merely compensating for "small errors in late sub-trajectories."
> >
> > In summary, CMQuant's novelty lies in both the individual modules (CMs-specific innovations) and the integrated framework (synergistic solution to quantization's core contradictions in CMs). We hope this clarification addresses your concern, and we are happy to provide further details if needed.

---

> > > ### Author Response · Authors · 2025-11-19
> > > **Rebuttal by Authors [3/3]**
> > >
> > > **2. Concerns in the experiment**
> > >
> > > Considering the article length and ease of comparison, we have included all existing PTQ and QAT methods suitable for Consistency Models (CMs) in Tables 1 and 2. To demonstrate the effectiveness of our approach, we also incorporated existing PTQ methods such as SVDQuant into the tables. Our key finding, which we aim to present through these comparisons, is that while existing PTQ methods perform well on full-iteration diffusion models, they yield poor results when applied to CMs. For existing QAT methods, we have adapted them to CMs ourselves, and these adapted QAT methods serve as our primary baselines for comparison.
> > >
> > > We sincerely apologize for any confusion caused. Your suggestion is highly valuable, and we will separate the experiments related to PTQ and QAT to make the tables more intuitive and clear, thereby avoiding misunderstandings. Additionally, we have supplemented comparative experiments with the latest methods, evaluating the Frechet Inception Distance (FID, lower is better) on the SDXL model using the MS-COCO-10K dataset under the configuration of Classifier-Free Guidance (CFG) = 1.
> > >
> > > | Method       | FID↓  | Method          | FID↓  | Method         | FID↓  |
> > > |--------------|-------|-----------------|-------|----------------|-------|
> > > | PCM + QERA   | 47.12 | PCM + Caldera   | 48.15 | PCM + LQ-LoRA  | 46.44 |
> > > | TCD + QERA   | 32.09 | TCD + Caldera   | 33.06 | TCD + LQ-LoRA  | 33.14 |
> > > | TDD + QERA   | 33.51 | TDD + Caldera   | 36.21 | TDD + LQ-LoRA  | 34.33 |
> > > | Ours         | 30.56 |                 |       |                |       |
> > >
> > > **3. About activation quantization**
> > >
> > > As explained in the experimental section of the paper, our method quantizes both activations and weights to **INT4**. As mentioned earlier, HGS-LoRA is not a weight initialization compensation method, and we only use asymmetric quantization for activations and do not impose additional restrictions on them.
> > >
> > > Regarding the reviewer’s comment on the claim "most existing methods focus solely on ...", we fully agree with your insight. There is indeed extensive research on activation quantization in the broader quantization field. However, this statement is specifically made in the "Quantization-Aware Training with LoRA" section of our paper, referring exclusively to LoRA-involved quantization methods. Through our in-depth investigation, we found that most of these methods do not quantize activations. Specifically, CALDERA, LQ-LoRA, and QERA only perform quantization on Weights and do not involve any quantization of activations. We apologize for the ambiguity in the original expression and will clarify this specific scope in the revised manuscript to avoid misunderstandings.
> > >
> > > We hope these revisions have addressed your concerns and enhanced the paper’s readability, and we are grateful for your guidance in helping us refine the work. Please feel free to let us know if you have any further questions or suggestions. **We look forward to your feedback.**

---

> > > > ### Comment · Reviewer_ECwp · 2025-11-19
> > > >
> > > > > About activation quantization
> > > >
> > > > If you also quantize the activation, the layer output error becomes $|| W_q x_q - W x ||_2^2$ instead of $|| W_q x - W x ||_2^2$.    Could you justify your objective in Eq(6) which keeps the activation un-quantized?
> > > >
> > > > My concern is that for aggressive quantization like INT4, quantizing activation to INT4 usually does more harm to the model performance compared to INT4 weights because of activation outliers ([Outlier Suppression+](https://arxiv.org/abs/2304.09145), [llm.int8](https://arxiv.org/pdf/2208.07339)).

---

> ### Comment · Reviewer_ECwp · 2025-11-19
>
> > **1.2 HGS-LoRA vs. QERA / CALDERA**
>
> - You may interpret this activation scaled SVD from a different perspective, but **CMQuant's optimization objective (Eq (6) in Appendix A2), and solution (Eq (10))** takes the same form as the works published before. They are essentially the same problem & solution.
>
>   More precisely, previous works minimize the layer output error $E|| Wx - W_q x||_2^2 $ or $||WX-W_qX||_2^2$, which is the same as your objective in Eq(6) Appendix A2.
>
>   and they define the scaling matrix as
>   - $E[xx^T]$ in [QERA](https://arxiv.org/abs/2410.06040) (see their eq(11), where $x$ denotes an activation vector
>   - $XX^T$ in [SVD-LLM](https://arxiv.org/pdf/2403.07378) (see their eq(3)) and [CALDERA](https://arxiv.org/pdf/2405.18886)'s Lemma 4.2
>
>   CMQuant defines $H := xx^T$, i.e., only sampling a single input, which may be questionable (only one sample seems not enough?) but essentially the same as these works. Is my understanding correct?
> -  Yes, apart from error compensation, this scaled SVD method can be used to **initialize the trainable low-rank terms** in qLoRA, but this is **also explored in QERA (See their Section 4.2) and CALDERA (See their Section 5.2)**.

---

> ### Author Response · Authors · 2025-11-20
> **Rebuttal by CMQuant Authors**
>
> Dear Reviewer,
>
> Thank you for your insightful comments. We have analyzed the issues raised and provide the following responses:
>
> **1. Regarding the loss function and activation change measurement:**
>
>
>   - The loss $||Wx - W_qx||^2$ is widely used in quantization compression and is not an innovative point proposed by us; it has been adopted in numerous existing works.
>
>
>   - Using $XX^T$ to measure activation changes first appeared in the OBS [1] paper (the predecessor of the well-known GPTQ[2]), and it is not exclusive to SVD-LLM or CALDERA. We apologize for any misunderstanding caused by our formula expression. Capturing activation changes in this way typically requires calibration with a dataset, which is also the case for CMQuant.
>
> [1] OBS:Optimal Brain Surgeon and General Network Pruning
>
> [2] GPTQ: Accurate Post-Training Quantization for Generative Pre-trained Transformers
>
> **2. The key differences between our method and previous works are as follows:**
>
>   - CMQuant: Decomposes the **original pre-trained weights**, freezes the principal components that are susceptible to quantization effects, and further decomposes the remaining components as the initialization weights for LoRA.
>
>   - QERA/CALDERA: Focus on optimizing quantization parameters through error minimization, decomposing the weight quantization error matrix $||W - W_q||$ to serve as the initial A and B matrices of LoRA.
>
>   - Beyond differences in core objectives, SVD decomposition targets, and initialization methods, CMQuant also differs from existing works in its application domain. QERA, CALDERA, and SVD-LLM are all excellent works that utilize SVD for compression, but like CMQuant, each has its own focus and applicable scenarios.
>
> **3.Regarding activation quantization:**
>
> - In the initial experimental phase of the paper, we only intended to perform weight quantization for the CM model, hence the use of the current formula. When we found that weight-only quantization had a minor impact on the accuracy of CMQuant, we further implemented activation quantization. We did experiment with $||Wx - W_qx_q||^2$, but the training oscillated severely in the early stages, making it prone to non-convergence, and the performance was worse than the version that does not consider activation quantization error. We attribute this to the fact that activation quantization error is significantly affected by data distribution and inter-layer heterogeneity. When this error is not considered, SVD focuses solely on the low-rank structure of the weights themselves, leading to more stable singular value ordering. Especially under high compression ratios, it can better preserve the core capabilities of the model and enhance the generalization of the method.
>
> - Benefiting from the fact that CMQuant is a training-based quantization method, it exhibits a certain degree of robustness to the impact of activation outliers on quantization. Nevertheless, considering activation quantization does increase the training difficulty and slow down the convergence speed. After incorporating activation quantization, compared to weight-only quantization, we not only reduced the learning rate but also increased the number of training iterations and screened training samples. We also adopted a progressive training approach: first training with INT8 activation as the target, then switching to INT4.
>
> The reviewer's professional acumen and extensive knowledge are highly admirable. We will revise the related work section to include discussions on QERA, CALDERA, and SVD-LLM. Additionally, we will add a comparative analysis and discussion on whether to consider activation quantization in the appendix. **And code will be released.**
>
> In summary, we sincerely thank the reviewer for the careful review, which has helped clarify the innovative points of our paper and improve the accuracy of its expression. **We will include a separate section to acknowledge your contributions.**

---

> > ### Comment · Reviewer_ECwp · 2025-11-27
> >
> > Just for clarification, I didn't mean using $XX^T$ or the objective $ ||Wx - W_qx||^2 $ is exclusive to SVD-LLM or CALDERA. I mean one of your contributions (the solution in Eq (5)) has been explored in previous works like SVD-LLM and Caldera.

---

> ### Author Response · Authors · 2025-11-27
> **Thank you once again for your time and effort in reviewing our work ! !**
>
> Dear Reviewer ECwp,
>
> We fully agree with your comment. Both our work and previous studies explore the same error formula, sharing a consistent research focus on this key mathematical foundation.
>
> As detailed in Section 2 of our prior rebuttal （**''The key differences between our method and previous works are as follows:''**), we would like to further clarify the core distinction:
>
> - **Different initialization strategy**:
>   CMQuant decomposes the original pre-trained weights first, then freezes the principal components that are susceptible to quantization effects. The remaining components are further decomposed to serve as the initialization weights for LoRA.
>
> - **Application domain and Distinct Objective**:
>   Beyond the differences in SVD decomposition targets and initialization methods, CMQuant also diverges from existing works in its specific application domain. The purpose of our design is to strike a balance between preserving the original pre-trained weights and enabling targeted fine-tuning in CM, thereby guiding the model to explore a quantization-friendly direction.
>
> If you have any remaining comments, additional feedback, or further points that require clarification, please do not hesitate to let us know. Your insights are invaluable to refining our work, and we are committed to addressing any outstanding issues to enhance the quality of the paper. **We sincerely appreciate your diligence and sense of responsibility.**
>
> Best regards.
>
> The Authors of CMQuant

---

### Official Review · Reviewer_7FYm · 2025-10-31

**Soundness:** 3
**Presentation:** 3
**Contribution:** 3
**Rating:** 6
**Confidence:** 3

**Summary:**

This paper tackles low-bit quantization of Consistency Models (CMs) and proposes CMQuant, a single-stage training scheme that jointly performs CM distillation and quantization-aware LoRA adaptation. The key idea is a trajectory-aware objective (TDPT) that aligns early steps to a frozen, untrained quantized copy of the CM to anchor the student in the quantization space, and aligns late steps to the FP16 CM to preserve endpoint quality. The parameterization uses HGS-LoRA, which splits weights into quantization-sensitive principal components (frozen and quantized) and a trainable LoRA on the residual subspace, and QRA to adapt LoRA rank per layer under a fixed parameter budget using activation dynamics and spectral cues. Empirically, the method reports the first W4A4 (weights and activations) CM with minimal quality regression and tangible speed/memory gains over PTQ and QAT baselines on SDXL/FLUX-style CMs.

**Strengths:**

1. Addresses a real gap—W4A4 quantization for CMs with multi-step error compounding.
2. Conceptual insight: using a frozen, untrained quantized teacher early provides a “low-bit anchor” rather than redundant self-copying, improving stability across steps.
3. Strong empirical results: maintains generation quality while enabling low-bit inference (reported W4A4) with measurable throughput and memory benefits.

**Weaknesses:**

1. The paper does not explicitly state whether LoRA is merged into the base and re-quantized for inference; the W4A4 claim implies it, but the procedure should be made explicit.
2.  QLoRA/QALoRA ranks (r) and parameter budgets are not reported in experiments; fairness may be affected without matching total LoRA capacity/compute.
3. QRA specifics: criteria, thresholds, and sensitivity are heuristic in description; needs ablation vs fixed-r, and analysis of stability/sensitivity to the allocation policy.
4. The speed evaluation primarily compares against the 20-step FP16 diffusion teacher rather than a like-for-like 4-step FP16 CM baseline, conflating fewer-step sampling with low-bit kernel effects and potentially overstating the acceleration attributable to quantization itself.

Minor Weaknesses
a）Inconsistent terminology/acronyms: the paper alternates between “Trajectory Distillation with Phased Targets (TDPT)” and “Trajectory Distillation with Adaptive Targets (TDAT)”
b) The main text cites “Appendix A.4” for the single-step error discussion, but the corresponding section in the appendix is mislabeled as a second “A.3”

**Questions:**

1.  Do you merge the learned LoRA into the base weights and then re-quantize for deployment? If not, how is W4A4 preserved without a high-precision branch?
2. What LoRA ranks (r) and total parameter budgets were used for QLoRA/QALoRA? Can you report sensitivity to r and ensure matched budgets/compute?
3. HGS-LoRA details: how is the Hessian approximated (e.g., Hutchinson/K-FAC/diag Fisher), and how is the principal subspace/rank k chosen per layer?
4. TDPT schedule: how are early vs late steps defined and weighted? Is the split static or adaptive? Ablate the split point and the loss weights; compare to using only FP teacher vs only quantized teacher.
5. Could you add a direct comparison against an FP16 CM (same architecture, same number of steps/resolution/batch) in terms of latency and throughput to isolate the net gains from low-bit quantization? Also, please clarify whether inference actually uses W4A4 kernels with activation quantization enabled, and specify the exact hardware, libraries/kernels, and batch settings used.

---

> ### Author Response · Authors · 2025-11-13
> **Rebuttal by Authors [1/2]**
>
> Dear Reviewer,
>
> We sincerely thank you for your valuable time and efforts in reviewing our manuscript. We have addressed each comment and made the necessary revisions to improve the quality and clarity of our manuscript.
>
> **1. Quantization Method**
>
> We sincerely apologize for the confusion caused by our lack of detailed and explicit explanation. To clarify: our method merges LoRA weights into the base model first, then quantizes both the weights and activations to W4A4.
>
> **2. Concerns in LORA Rank**
>
> Regarding the LoRA Rank $r$: To ensure fair comparison with PCM, TCD, and their variants, our method retains their default rank setting of 128—this consistency applies to both QLoRA and QALoRA.
>
> For our proposed approach QRA, as detailed in Algorithm 1 of the appendix, we adopt  $1.5r$ and $0.5r$ for different layers while keeping the total number of parameters unchanged. The shapes of the LoRA matrices A and B are $d\*r$ and $r\*d$, where $d$ represents the output dim of weight. For the same $d$, our QRA method makes layers with $1.5r$ account for 10% and layers with $0.5r$ account for 10%, thereby ensuring that the total number of parameters remains unchanged.
>
> This design is grounded in our key findings: layers exhibit significant differences in rank, temporal variation of inputs, and weight convergence speed. We observed that increasing the rank for layers with training-time weight oscillations, slow convergence, or high activation variability accelerates model convergence, whereas reducing the rank for other layers has minimal impact on performance—this forms the basis of our third method. In experiments, we found 1.5 and 0.5 to be reasonable scaling factors. While finer-grained parameter tuning could potentially yield better results, we opted for fixed ratios to avoid increasing total parameters, thereby highlighting the effectiveness of our method rather than performance gains from additional parameters.
>
> On the MS-COCO-5K dataset with CFG=1, we conducted ablation experiments on $r$ using SDXL. We found that a smaller $r$ (e.g., 32) significantly degrades the model performance, while a larger $r$ improves the model performance but introduces more parameters. However, when $r$ reaches a certain value, the accuracy gain of the model decreases significantly. The ablation experiments in Table 9 of the paper also demonstrate the effectiveness of QRA.
>
> | r    | 32    | 64    | 96    | 128   | 160   | 192   | 224   |
> | :--- | :---- | :---- | :---- | :---- | :---- | :---- | :---- |
> | Ours_method (FID⬇) | 87.87 | 41.42 | 36.87 | 34.16 | 33.36 | 32.68 | 32.66 |
>
> **3. HGS-LoRA details**
>
> For the Hessian approximation, we refer to the $X*X^T$ method, which is also the most commonly used one, as in GPTQ.
>
> Regarding the parameter k: To ensure fair comparison with PCM, TCD, and their variants, our method retains their default LoRA rank setting of 128. Therefore, in HGS-LoRA's principal component extraction, we also set (k = 128) to preserve the principal components of the original model as much as possible. Your question is thought-provoking. In future work, we plan to investigate how to dynamically determine the value of k through theoretical formulations to achieve better performance. We are readily prepared to provide more detailed ablation experiments if needed.
>
> **4. TDPT schedule**
>
> Regarding the design of the TDPT schedule, the comparison between using only an FP teacher and using only a quantized teacher is equivalent to the selection of T' in Equation 4 of the paper. Regarding the ablation experiments on the parameter T' in the TDPT method: We conducted experiments using SDXL on the MS-COCO-10K dataset, which revealed that setting T' = T/4 yields favorable results. While we believe that more fine-grained parameter tuning could potentially further improve performance, T/4 already represents a robust and effective choice for practical use.
>
> | T'          | T/20   | T/10 | T/5  | T/4  | T/2  | T*2/5 | T*4/5 |
> |-------------|----------|--------|--------|--------|--------|-----------|-----------|
> | SDXL (FID ↓) | 31.50  | 31.19  | 30.87  | 30.56  | 32.56  | 33.12     | 33.56     |

---

> ### Author Response · Authors · 2025-11-13
> **Rebuttal by Authors [2/2]**
>
> **5. W4A4 Inference with Activation Quantization Enabled**
>
> Our inference pipeline fully utilizes W4A4 (4-bit weights and 4-bit activations) kernels with activation quantization enabled, based on the QuaRot implementation [1]. Specifically:
>
> - **Weight quantization**: Weights are quantized to 4-bit using asymmetric quantization with per-group scaling (group size = 128), following the packing format where two 4-bit values are packed into a single uint8 byte to minimize memory overhead.
>
> - **Activation quantization**: Activations are dynamically quantized to 4-bit during inference (on-the-fly) using symmetric quantization, with scaling parameters precomputed during calibration on a representative dataset. This ensures that both weights and activations are represented in 4-bit precision when fed into the core matrix multiplication (GEMM) kernels.
>
> - **End-to-end validation**: We verified via tensor type inspection (e.g., torch.Tensor.dtype for intermediate activations) that quantized activations (4-bit) are indeed used in the GEMM operations, rather than falling back to higher precision (e.g., 8-bit or FP16).
>
> - **Dependencies**: PyTorch 2.0.1 (for tensor operations and autograd), CUDA 11.8 (for kernel compilation), and cuDNN 8.7 (for auxiliary GPU-accelerated functions).
>
> All experiments were conducted on a single NVIDIA A100-SXM4-80GB GPU, based on the Ampere architecture (equipped with 6912 CUDA cores and 432 Tensor Cores). This hardware natively supports INT4/FP16 mixed-precision computations through its Tensor Cores, making it well-suited for efficient execution of 4-bit quantized kernels. All inference runs were performed with a batch size of 1 to align with typical single-sample generation scenarios.
>
> Our method is the first to integrate both consistency models (CMs) and quantization. For methods that solely involve CMs, our approach—by incorporating quantization—achieves significant improvements in both memory efficiency and generation speed. For quantization-only methods, our integration of CMs (which reduces the number of inference iterations) provides a distinct advantage: even when compared to other 4-bit activation/weight quantization methods, our approach is substantially faster due to fewer required iterations. A direct comparison against only one type of method would be unfair. Additionally, speed testing is a time-consuming task that requires implementing Triton and CUDA operators, and relevant code is not available for some methods.
>
> | Model | Method                | img/sec | Speedup Ratio |
> | :---- | :-------------------- | :------ | :------------ |
> | FLUX  | FP16 (20 steps)       | 74.33   | 1.0×          |
> |       | SVDQuant W4A16 (20 steps) | 46.52   | 1.59×         |
> |       | CMQuant W4A4 (20 steps)   | 48.50   | 1.53×         |
> |       | CMQuant W4A4 (4 steps)    | 21.68   | 3.43×         |
>
> Your concern is quite reasonable. To alleviate your worries, we conducted tests on other methods with 20 steps and SVDQuant. It was found that under the same number of steps, even with A16, the SVDQuant method is slightly faster than ours. This may be due to the elegant implementation of their computation kernels. However, it still lags far behind our 4-step method.
>
> **About writing**
>
>  We are deeply sorry for these editorial errors pointed out by the reviewer. This was a negligence in our work, and we sincerely apologize for the inconvenience caused. These issues likely arose from an incorrect submission of the final version of the manuscript, but we rectified them in the updated version:  We have standardized the naming of Trajectory Distillation with Phased Targets (TDPT) and corrected Appendix A.4.
>
> We hope these revisions have addressed your concerns and enhanced the paper’s readability, and we are grateful for your guidance in helping us refine the work. Please feel free to let us know if you have any further questions or suggestions. **We look forward to your feedback.**
>
> [1] QuaRot: Outlier-free 4-bit inference in rotated LLMs

---

### Official Review · Reviewer_wRLa · 2025-10-31

**Soundness:** 2
**Presentation:** 2
**Contribution:** 2
**Rating:** 4
**Confidence:** 2

**Summary:**

This paper proposes a quantization-aware PEFT framework named CMQuant for CMs in W4A4. The authors identify two main challenges when applying quantization to CMs: (1) quantization errors accumulate and amplify across inference iterations due to the trajectory fitting nature of CMs, and (2) existing LoRA-based training is incompatible with quantization as it disrupts initialization consistency. Three contributions, TDPT, HGS-LoRA and QRA, are proposed to address the problems, and experimental results show improvements in throughput and memory savings.

**Strengths:**

* Applying quantization to CMs is an interesting and novel problem.

**Weaknesses:**

* It feels like this work is stitching three pieces together, making novelty a bit weak.
* Writing could be improved. For example, Figures 1 and 2 are not mentioned in the text, making the motivation confusing.
* How do TDPT, HGS-LoRA, and QRA interact and affect the results?
* Table 6 misses comparison with other related works.
* The datasets used are limited - how about evaluation on other datasets (e.g., ImageNet)?

**Questions:**

* In Figure 1 and Table 1, CMQuant seems to be close to TCD. But TCD is missing in Table 2. Could the authors explain why?
* Could the authors provide justifications for the selection of parameters (e.g., T' and r)? How sensitive are they?

---

> ### Author Response · Authors · 2025-11-13
> **Rebuttal by Authors [1/2]**
>
> Dear Reviewer,
>
> We sincerely thank you for your valuable time and efforts in reviewing our manuscript. We have addressed each comment and made the necessary revisions to improve the quality and clarity of our manuscript.
>
> **1. Concerns in TCD**
>
>  In practice, our main comparison baselines are PCM, TCD, TDD and their variants combined with quantization methods. It is worth noting that the official release of TCD only includes weights for the SD series. We attempted to train a floating-point version of TCD for the FLUX series ourselves, but the training results were far from satisfactory and significantly lagged behind the performance of SDXL. This may suggests that the TCD method may not be well-suited for FLUX-series models.
>
> **2. Selection of parameters**
> - Regarding the ablation experiments on the parameter T' in the TDPT method: We conducted experiments using SDXL on the MS-COCO-10K dataset, which revealed that setting T' = T/4 yields favorable results. While we believe that more fine-grained parameter tuning could potentially further improve performance, T/4 already represents a robust and effective choice for practical use
>
>     | T'          | T/20   | T/10 | T/5  | T/4  | T/2  | T*2/5 | T*4/5 |
>     |-------------|----------|--------|--------|--------|--------|-----------|-----------|
>     | SDXL (FID ↓) | 31.50  | 31.19  | 30.87  | 30.56  | 32.56  | 33.12     | 33.56     |
> - Regarding the parameter $r$: To ensure fair comparison with PCM, TCD, and their variants, our method retains their default LoRA rank setting of 128. Therefore, in HGS-LoRA's principal component extraction, we also set $r = 128$ to preserve the principal components of the original model as much as possible.Your suggestion is highly constructive. In future work, we plan to investigate how to dynamically determine the value of r through theoretical formulations to achieve better performance. We are readily prepared to provide more detailed ablation experiments if needed.
>
> **3. How TDPT, HGS-LoRA, and QRA interact and affect**
>
> The three core modules of CMQuant (TDPT, HGS-LoRA, and QRA) are not simply concatenated; instead, they form a progressive collaborative architecture centered on "resolving the core contradictions of 4-bit quantization for Consistency Models (CMs)."
> - **Across iterations**: TDPT blocks error accumulation, providing a stable optimization foundation for HGS-LoRA and QRA.
> - **Within a single iteration**: HGS-LoRA and QRA operate from the perspectives of initialization and resource allocation, respectively. HGS-LoRA guides optimization directions, while QRA allocates resources to critical layers, amplifying their synergistic effect.
>
> These three modules are logically deeply integrated, collectively achieving the goal of "4-bit quantization without performance loss." Existing quantization methods (e.g., QLoRA, SVDQuant) are designed for large language models or conventional diffusion models and cannot address the quantization challenges arising from CMs' unique properties. The innovations in CMQuant's modules are all tailored to CMs' characteristics:
> - **TDPT** ：It alleviates the "error accumulation and amplification" issue in CMs quantization across iterations by proposing a "staged target assignment" strategy: early sub-trajectories adopt outputs from untrained quantized models as targets (ensuring self-consistency within the quantized space and avoiding cross-space fitting errors), while late sub-trajectories use outputs from floating-point models as targets (to align with high-precision generation objectives). This marks the first quantization-friendly distillation method tailored to the iterative inference characteristics of CMs, blocking error accumulation at its source.
>
>     Without TDPT, quantization errors would accumulate exponentially during CMs iterations. Even if HGS-LoRA optimizes LoRA initialization, training would diverge due to excessive input errors. Through its staged target assignment, TDPT confines errors in early sub-trajectories within the quantized space, thereby reducing HGS-LoRA’s optimization burden to merely compensating for "small errors in late sub-trajectories."
>
> - **HGS-LoRA**: Within a single iteration, it uses the Hessian matrix of activations to measure the contribution of weights to quantization errors, accurately identifying and freezing "quantization-sensitive principal components"; meanwhile, it initializes LoRA weights with "quantization-insensitive non-principal components," guiding model updates along "quantization-friendly directions." Without HGS-LoRA, ordinary LoRA initialization would cause quantized weight distributions to deviate from pre-trained states. Even if TDPT controls error sources, the model might oscillate or fail to converge due to incorrect optimization directions.

---

> ### Author Response · Authors · 2025-11-13
> **Rebuttal by Authors [2/2]**
>
> - **QRA**: Conventional LoRA assigns fixed ranks to all layers, ignoring a key characteristic of CMs—significant differences in activation fluctuations, weight distributions, and training convergence speeds across layers. Based on "activation iterative fluctuation variance" and "weight singular value variance," QRA allocates higher ranks to highly sensitive layers to ensure expressive capacity, and lower ranks to low-sensitive layers to save resources. Without increasing total parameters, it precisely directs limited LoRA resources to "layers most affected by quantization errors," further improving model performance.
>
>
>
> In summary, CMQuant's novelty lies in both the individual modules (CMs-specific innovations) and the integrated framework (synergistic solution to quantization's core contradictions in CMs). We hope this clarification addresses your concern, and we are happy to provide further details if needed.
>
> **4. Concern about other details**
>
> - **Evaluation datasets**: Consistent with other related works, we adopt the same evaluation datasets. Specifically, the COCO dataset is utilized, which is representative in the text-to-image generation field.
>
> - **Table 6 Explanation**: Our method is the first to integrate both consistency models (CMs) and quantization. For methods that solely involve CMs, our approach—by incorporating quantization—achieves significant improvements in both memory efficiency and generation speed. For quantization-only methods, our integration of CMs (which reduces the number of inference iterations) provides a distinct advantage: even when compared to other 4-bit activation/weight quantization methods, our approach is substantially faster due to fewer required iterations. A direct comparison against only one type of method would be unfair. Additionally, speed testing is a time-consuming task that requires implementing Triton and CUDA operators, and relevant code is not available for some methods.
>
>     | Model | Method                | img/sec | Speedup Ratio |
>     | :---- | :-------------------- | :------ | :------------ |
>     | FLUX  | FP16 (20 steps)       | 74.33   | 1.0×          |
>     |       | SVDQuant W4A16 (20 steps) | 46.52   | 1.59×         |
>     |       | CMQuant W4A4 (20 steps)   | 48.50   | 1.53×         |
>     |       | CMQuant W4A4 (4 steps)    | 21.68   | 3.43×         |
>
>     Your concern is quite reasonable. To alleviate your worries, we conducted tests on other methods with 20 steps and SVDQuant. It was found that under the same number of steps, even with A16, the SVDQuant method is slightly faster than ours. This may be due to the elegant implementation of their computation kernels. However, it still lags far behind our 4-step method.
>
> - **About writing**: In fact, Figure 1 is mentioned in the introduction. We have fully incorporated your suggestion by revising the relevant sections introducing the paper’s motivation, integrating both Figure 1 and Figure 2 into this part. Additionally, we have reviewed the entire manuscript multiple times to ensure all expressions are accurate and formatting is standardized, while enhancing the paper’s readability and comprehensibility.
>
> We hope these revisions have addressed your concerns and enhanced the paper’s readability, and we are grateful for your guidance in helping us refine the work. Please feel free to let us know if you have any further questions or suggestions. **We look forward to your feedback.**

---

> ### Author Response · Authors · 2025-11-27
> **Thank you once again for your time and effort in reviewing our work. We respectfully invite you to review our Rebuttal.**
>
> Dear Reviewer wRLa,
>
> I hope this message finds you well. As the rebuttal discussion period is drawing to a close (with less than one week remaining), we wanted to follow up to confirm whether all your concerns have been adequately addressed in our previous response.
>
> If you have any remaining comments, additional feedback, or further points that require clarification, please do not hesitate to let us know. Your insights are invaluable to refining our work, and we are committed to addressing any outstanding issues to enhance the quality of the paper.
>
> Thank you sincerely for your time, diligence, and constructive input during the review process. We greatly appreciate your efforts in helping us improve the manuscript.
>
> Best regards,
>
> CMQuant Authors

---

### Meta-Review · Area_Chair_RzDY · 2025-12-22

**Summary:**

After carefully reading the rebuttal and paper, I don't think the authors' rebuttal successfully addresses the concerns. While this paper is interesting in addressing W4A4 quantization of CMs, the contribution is not very strong. Moreover, I think it would be better to provide more systematic strategies to calibrate the hyperparameters and consolidate generalization to new models/architectures. I am inclined to reject this submission. I encourage the authors to take into account this feedback for a further submission.

**Reviewer Concerns:**

1. Reviewer 7FYm  (partly addressd). TDPT schedule and SVDQuant comparison somewhat weaken  the contribution of this submission.
2. Reviewer ECwp (partly addressed).  The reviewer provided further feedback to the rebuttal and still had some concerns about contribution, some important ideas had been explored by previous work.
3. Reviewer wRLa (partly addressed).  Unsoved concerns: Missing TCD in Table 2, selection of parameters (a systemtic way is preferred) and comparision (Table 6)

**Reviewer Scores:**

1. Reviewer 7FYm would keep 6 or decrease the score to 4.
2. Reviewer ECwp would keep 4.
3. Reviewer wRLa would keep 4.

---

### Decision · Program_Chairs · 2026-01-26

Reject